# FASTER KERNEL DENSITY ESTIMATION VIA HASHING BASED TIME-SPACE TRADEOFFS

## ABSTRACT

In this paper we study the Kernel Density Estimation (KDE) problem: Given a dataset $\mathcal{P}$ of $n$ points in Euclidean space and a kernel $K(p, q)$, prepare a low space data-structure that given a query $q$ can quickly output a $1 \pm \epsilon$ approximation to $\mu = (\sum_{p \in \mathcal{P}} K(p, q))/n$. Recent advances have used tools from Locality Sensitive Hashing (LSH) and Approximate Nearest Neighbor (ANN) search to build KDE data-structures with query time *sublinear* in $1/\mu$ and space linear in $1/\mu$, with Charikar et al. (2020) achieving the current best query time of $\approx 1/\mu^{0.173}$ for the popular Gaussian kernel. Our main result is a data-structure with significantly improved query time $\approx 1/\mu^{0.05}$, at the expense of somewhat higher space complexity of $\approx 1/\mu^{4.15}$. More generally, our techniques give the first known query time vs space tradeoffs for KDE: for any $\delta \geq 0$ we can design a KDE data-structure with space with $1/\mu^{1+\delta}$ dependence and query time with $1/\mu^{\xi(\delta)}$ dependence, where $\xi(\delta)$ is a non-increasing function of $\delta$. Importantly for the linear space regime, i.e $\delta = 0$, we obtain a query time of $1/\mu^{0.1865}$, improving the non-adaptive KDE bound from Charikar et al. (2020) and nearly matching the bound of Charikar et al. (2020) with a significantly simpler analysis.

## 1 INTRODUCTION

Kernel Density Estimation (KDE) is a fundamental and widely studied problem in statistical learning theory and artificial intelligence (Fan, 2018; Schölkopf & Smola, 2002; Joshi et al., 2011; Arias-Castro et al., 2016). Formally KDE is defined as follows - Given $\epsilon > 0$ and a dataset $\mathcal{P}$ of $n$ points $\mathbf{p}_1, \ldots, \mathbf{p}_n \in \mathbb{R}^d$, preprocess it into a small space data-structure that allows one to quickly approximate, given a query $\mathbf{q} \in \mathbb{R}^d$, the quantity

$$\mu^* = K(\mathcal{P}, \mathbf{q}) = \frac{1}{|\mathcal{P}|} \sum_{\mathbf{p} \in \mathcal{P}} K(\mathbf{p}, \mathbf{q}), \tag{1}$$

up to multiplicative $1 \pm \epsilon$ factor with probability 0.9, where the kernel function $K(\mathbf{p}, \mathbf{q})$ is a monotone decreasing function of $\|\mathbf{p} - \mathbf{q}\|$. The Gaussian kernel,

$$K(\mathbf{p}, \mathbf{q}) = e^{-\|p-q\|_2^2/(2\sigma^2)}, \tag{2}$$

is a important example of a kernel widely used and will be the main focus of our paper, although many others (eg. Laplace, exponential, polynomial) are also used sometimes used (Shawe-Taylor & Cristianini, 2004; Williams & Rasmussen, 2006). Moreover recent works have used fast Gaussian KDE primitives for speeding up attention computation in modern transformer based LLMs (Zandieh et al., 2023; Indyk et al., 2025).

Unfortunately the exact algorithm for this problem does a linear scan over $\mathcal{P}$ at query time and thus runs in time linear in $n$, making it not scalable for large datasets. Thus most practical algorithms resort to reporting approximate kernel density evaluation at query time. In the low dimensional regime tree based algorithms Greengard & Strain (1991); Gray & Moore (2001); Gan & Bailis (2017) give efficient approximations, however their running times are exponential in $d$ making them not scalable for high-dimensional datasets. In the rest of the paper, we use the notation $\mu^*$ defined as $\mu^* := K(\mathcal{P}, \mathbf{q})$ to denote the true kernel density for a query $\mathbf{q}$, and $\mu$ denotes a quantity that satisfies $\mu^* \leq \mu \leq 4\mu^*$, using standard techniques we can assume such a $\mu$ is known to us (see

Remark 3 Charikar et al. (2020)). In the high dimensional regime $d = \Omega(\log n)$ uniformly sampling $O(1/(\epsilon^2) \cdot 1/\mu)$ dataset points $\widetilde{\mathcal{P}}$ from $\mathcal{P}$ and reporting $K(\widetilde{\mathcal{P}}, \mathbf{q})$ at query time suffices to obtain a $1 \pm \epsilon$ estimate of the true kernel density $K(\mathcal{P}, \mathbf{q})$. The line of work initiated in the work of Charikar & Siminelakis (2017) improved upon random sampling by creating Gaussian KDE data-structures with *sublinear* in $1/\mu$ query time and linear in $1/\mu$ space. In the subsequent discussion all methods have polynomial in $d$ and $1/\epsilon$ dependence in the query time and space, so we suppress them for readability. Furthermore we use $\widetilde{O}(\cdot)$ to hide polynomial factors in $d$ and $\log(n, 1/\mu)$. Charikar & Siminelakis (2017) used *Locality sensitive hashing* (LSH) (Indyk & Motwani, 1998; Andoni & Indyk, 2008), a fundamental technique in the approximate nearest neighbor (ANN) literature, to design Gaussian KDE data-structures with query time with a $1/\sqrt{\mu}$ dependence. Following this a line of work had subsequent improvements using LSH culminating in the work of Charikar et al. (2020) which achieved a $1/\mu^{0.25}$ dependence in the query time using a data-independent LSH and $1/\mu^{0.173}$ using a much more involved data-dependent LSH. These approaches used *symmetric* LSH constructions, and our main contribution is to use advances in *asymmetric* LSH constructions (Andoni et al., 2017; Razenshteyn, 2017) to improve upon these works.

We first present an overview of our contributions followed by presenting our main ideas and techniques. Finally we end the section by discussing related work.

## 1.1 OUR CONTRIBUTIONS

Our first result, that obtains data-structures for Gaussian KDE problem (see problem setup in Equation 5) with significantly improved query time using asymmetric LSH, is as follows,

**Theorem 1.** *(Informal) There exists a data-structure for the Gaussian KDE problem with expected query time $\widetilde{O}((1/\epsilon^2) \cdot 1/\mu^{0.051})$ and space $\widetilde{O}((1/\epsilon^2) \cdot 1/\mu^{4.15})$. There also exists a data-structure for the Gaussian KDE with expected query time $\widetilde{O}((1/\epsilon^2) \cdot 1/\mu^{0.1865})$ and space $\widetilde{O}((1/\epsilon^2) \cdot 1/\mu)$.*

The formal version of the above theorem is presented in Theorem 17. Of course we obtain the improved query time of $1/\mu^{0.051}$ at the expense of polynomial in $1/\mu$ space, however the use of asymmetric LSH allows us to tradeoff the space and query time of our data-structure. Thus even for the linear space, i.e. with $1/\mu$ dependence in the space, we obtain a query time with $1/\mu^{0.1865}$ dependence on $1/\mu$ that beats the previous best bound of $1/\mu^{0.25}$ using non-adaptive schemes. It is slightly worse than the data-dependent scheme of Charikar et al. (2020), which achieved a $1/\mu^{0.173}$ dependence, however our scheme has the advantage of being much simpler. We also show a more general result that presents time-space tradeoffs for Gaussian KDE data structures, in the following

**Theorem 2.** *(Informal) For any $\delta \geq 0$ there exists a data-structure for the Gaussian KDE problem with query time $\widetilde{O}((1/\epsilon^2) \cdot 1/\mu^{\xi(\delta)})$ and space $\widetilde{O}((1/\epsilon^2) \cdot 1/\mu^{1+\delta})$ where $\xi(\delta)$ as a function of $\delta$ is presented in right figure in Figure 1.*

To the best of our knowledge, ours is the first such tradeoff for KDE, the formal version of the above theorem is presented in Theorem 16. We now describe the main techniques used to prove our results.

## 1.2 TECHNICAL OVERVIEW

Our query time vs space complexity tradeoffs for KDE are obtained by a novel instantiation of the framework of Charikar et al. (2020) that essentially reduces the KDE problem to a version of the Approximate Near Neighbor (ANN) problem. We thus start with an overview of that framework.

**KDE via (density constrained) approximate nearest neighbor search (ANN).** Charikar et al. (2020) reduce the problem of computing kernel density problem at a query $\mathbf{q}$ to logarithmic many approximate nearest neighbor (ANN) problems with the additional twist provided by *density constraints*. The main idea is to partition points $\mathbf{p} \in \mathcal{P}$ into a logarithmic number of distance scales according to the value of $K(\mathbf{p}, \mathbf{q})$, then estimate the number of points in each distance scale (i.e., at a certain Euclidean distance from $\mathbf{q}$), using approximate nearest neighbor search techniques such as Locality-Sensitive Hashing (LSH). Using standard scaling techniques, as in Charikar et al. (2020, Assumption 1 in Section 5), we conveniently re-write the Gaussian kernel for any point $\mathbf{p} \in \mathcal{P}$ as follows,

$$K(\mathbf{p}, \mathbf{q}) = \mu^{\|\mathbf{p}-\mathbf{q}\|_2^2},$$

and we denote $\mathcal{L}_j^{\mathbf{q}} \in \mathcal{P}$ denote the points in $\mathcal{P}$ with kernel value $K(p,q) \approx 2^{-j}$ for all values[1] of $j \in [0, J]$ for $J = \log(1/\mu)$. We denote the *distance scale* $x_j = j/J$, which thus conveniently allows us to write $\mathcal{L}_j^{\mathbf{q}} \in \mathcal{P}$ as all points with,

$$K(p,q) \approx \mu^{x_j} \quad \text{for } x_j \in [0,1].$$

See Section 3 for precise definitions. The framework of Charikar et al. (2020) randomly samples points in $\mathcal{P}$ at rate

$$p_j = (1/\mu)^{1-x_j} \cdot 1/n, \tag{3}$$

to create a subsampled dataset, then retrieves all point in $\mathcal{L}_j^{\mathbf{q}}$ surviving in this subsampled dataset using the symmetric LSH of Andoni & Indyk (2008).

Our work proposes to go beyond symmetric LSH to achieve the improvement, so it is more convenient to reformulate the Charikar et al. (2020) framework as applying a more general Approximate Near neighbor (ANN) data-structure. Recall that a $(c,r)$-ANN data-structure is an efficient data-structure that, assuming the existence of a point at distance at most $r$ from the query, returns a point at distance at most $cr$. When recovering points in $\mathcal{L}_j^{\mathbf{q}}$, i.e. at distance scale $x_j$, from the sampled dataset we invoke a $(c,r)$-ANN data-structure with the near radius $r$ corresponding to KDE contribution $\approx \mu^{x_j}$ and the far radius $cr$ corresponding to KDE contribution $\approx \mu$. We drop the subscript $j$ from scale $x_j$, since we will only work with scales.

**Remark 3.** Note that this classical guarantee that an $(c,r)$-ANN data-structure provides does not suit us, as we need to exactly retrieve all points at distance scale $\approx x$ from the sampled dataset, we will provide a new analysis of a powerful $(c,r)$-ANN data-structure that takes density constraints into account and achieves *exact* recovery efficiently.

**Exact recovery with approximate near neighbor search.** Charikar et al. (2020) use the symmetric LSH of Andoni & Indyk (2008) for this ANN problem, to provably recover points at distance scale $x \in [0,1]$ in sublinear time. The query time of this procedure is higher than that of the ANN problem because we need to retrieve point at exactly distance the near distance scale $x$ and during hashing, points at scale $y$ for $x < y \leq 1$ can collide with points at $x$, adding time needed in scanning and discarding these intermediate points. However this query time overhead can be controlled using *density constraints* - a simple Markov bound allows us to bound number of points at scale $y \in [0,1]$,

$$n(\mu)^{1-y} \ll n, \tag{4}$$

and furthermore it is unlikely that all such points collide with points at $x$. Charikar et al. (2020) bound the additional query time overhead by upper bounding the expected number of intermediate colliding points by multiplying density constraint upper bounds and with LSH collision probability of Andoni & Indyk (2008). This gives them the query time for recovering points at scale $x$ in the subsampled dataset for any fixed $x$. Summing this over log many possible values of $x$ they obtain a query time of $1/\mu^{0.25}$ up to log factors. Section 3 contains the precise details of this framework.

**Our contribution: query time reduction via asymmetric ANN.** Our main idea is to use the asymmetric LSH construction of Andoni et al. (2017) (see Section A) instead to recover points at scale $x$ from the subsampled dataset. For the $(c,r)$-ANN problem, this LSH allows us to design data-structures with space $n^{1+\rho_s+o(1)}$ and query time $n^{\rho_q+o(1)}$ for any $\rho_s, \rho_q \geq 0$ under the constraint,

$$(c^2+1)\sqrt{\rho_q} + (c^2-1)\sqrt{\rho_s} \geq 2c. \tag{5}$$

Choosing $\rho_s = \rho_q$ recovers the symmetric LSH of Andoni & Indyk (2008), but choosing it differently allows one to *tradeoff* lower query time for higher space for recovering points at scale $x$. This leads to an improvement over Charikar et al. (2020) because the maximum of query time in their reduction is achieved at a different distance scale $x \in [0,1]$ than the one that yields the space bound! Finding the best $\rho_s, \rho_q$ under constraint 5 for every $x \in [0,1]$ can be expressed as an optimization problem (see Section 4) and solved numerically (see Section 5). The exact optimum does not seem simple to obtain analytically, and we therefore resort to numerics. One interesting phenomenon emerges: unlike the $(c,r)$-ANN problem, which admits a solution with constant query time, the KDE tradeoffs that we achieve (see Fig. 1) do not yield a constant query solution. We next analytically show that this is not possible with present near neighbor search technology – an exciting open problem is to either prove a formal lower bound ruling out constant query KDE in polynomial space or bypass the inherent barrier in our scheme to get a KDE data structure with constant query time.

---

[1]More precisely, the set $\mathcal{L}_j^{\mathbf{q}}$ for $J = \log(1/\mu)$ is defined to capture all points with kernel value $K(\mathbf{p}, \mathbf{q}) = O(\mu)$ – the contribution of these points can be very easily estimated from a small sample.

**Why constant query KDE is not possible with known ANN results.** For a fixed scale $x \in [0, 1]$ the natural choice of the query exponent $\rho_q$ is to set it to 0 to ensure that at least the expected number of points colliding from the last scale $y = 1$, i.e. points at far distance $cr$ with kernel value $\approx \mu$, is at most $n^{o(1)}$. As otherwise any higher $\rho_q$ will lead to non-negligible contribution of points at far distance $cr$, as the $(c, r)$-ANN problem will have a non-negligible query time. Thus $\rho_q = 0$ is the natural choice, however again the overall query time will be higher than that for the $(c, r)$-ANN problem because of collisions from points at intermediate scales $y$ for $x < y \leq 1$. We now give a high level overview of this additional overhead. Fix an $x \in [0, 1]$ and recall from Equation 3 that first the dataset $\mathcal{P}$ is subsampled at rate $(1/\mu)^{1-x} \cdot 1/n$, leading to expected dataset size $(1/\mu)^{1-x}$. If we construct an asymmetric LSH for dataset size $(1/\mu)^{1-x}$ and $\rho_q = 0$, the probability for a point $p$ at scale $y$ for $x < y \leq 1$ to be scanned during query time turns out to be,

$$\left(\frac{1}{\mu}\right)^{-\left(\frac{(y-x)^2}{y(1-x)}\right)+o(1)}. \tag{6}$$

From density constraints 4, number points at scale $y$ is at most $n \cdot (\mu)^{1-y}$, which after subsampling gets reduced to $(1/\mu)^{y-x}$ in expectation. Thus overall the additional overhead due to points at scale $y$ is $(1/\mu)^{y-x}$ times the bound in Equation 6, and since there only log many values of $y$ to consider between $[x, 1]$ the overall overhead in query time is the following up to log factors,

$$\max_{y \in [x,1]} \left(\frac{1}{\mu}\right)^{(y-x)-\left(\frac{(y-x)^2}{y(1-x)}\right)+o(1)}, \tag{7}$$

In the expression above for $y = x$ and $y = 1$ the exponent is $o(1)$, however near $y = x$ the first linear term $y - x$ grows faster than the second term behaving roughly quadratically as $(y-x)^2$. Thus for any fixed $x \in [0, 1]$ the maximum happens for some point inside the interval $[x, 1]$. Furthermore since we need to recover points at logarithmic many scales $x \in [0, 1]$, the overall query time of this KDE data-structure is max of the above over all $x \in [0, 1]$, which using numerical methods is approximately $(1/\mu)^{0.09}$. This in general conveys the fact that even using this asymmetric LSH for query exponent $\rho_q = 0$ for all $x \in [0, 1]$, one cannot obtain arbitrarily small constant query time exponent at the expense of arbitrarily large polynomial space. However we can obtain a slightly better constant query time exponent than $0.09$ by optimizing setting $\rho_q$ for all $x \in [0, 1]$. For any $x \in [0, 1]$ and a general $\rho_q \geq 0$, Equation 6 is as follows,

$$\left(\frac{1}{\mu}\right)^{(1-x)\left(\rho_q - \frac{x}{y(1-x)^2}\left(\frac{y-x}{\sqrt{x}} - (y-1)\sqrt{\rho_q}\right)^2\right)+o(1)},$$

thus for a fixed $x \in [0, 1]$ the overall query time by optimizing over valid ranges of $\rho_q$ is as follows,

$$\min_{\text{valid } \rho_q} \max_{y \in [x,1]} \left(\frac{1}{\mu}\right)^{(y-x)+(1-x)\left(\rho_q - \frac{x}{y(1-x)^2}\left(\frac{y-x}{\sqrt{x}} - (y-1)\sqrt{\rho_q}\right)^2\right)+o(1)}$$

Finally the overall query time of our KDE data-structure is then the max of the above over all $x \in [0, 1]$. Solving this optimization problem leads to a query time roughly $(1/\mu)^{0.05}$. The precise details of this parameter setting and the optimization formulation are in Section 4.

**Query time for space** $1/\mu$. Obviously the space of the data-structure described previously is polynomial in $1/\mu$, roughly $1/\mu^4$, thus making it incomparable with previous works that had space at most $1/\mu$. However since the asymmetric LSH allows us to flexibly set either the space or query exponents for each recovery problems, we can carefully choose the space exponent so that the overall space of our data-structure to be at most $1/\mu$. This restricts the choice of the query exponent for each recovery problem as per Equation 5 leading to a higher query time. Overall this results in a data independent KDE data-structure with space $1/\mu$ and query time $1/\mu^{0.1865}$, which improves over the data independent bound of $1/\mu^{0.25}$ of Charikar et al. (2020). Moreover the query exponent is within $0.02$ of the exponent of the data dependent data-structure of the work of Charikar et al. (2020), which achieves a query time $1/\mu^{0.173}$, however our analysis is arguably much simpler. In general, our construction allows one to smoothly tradeoff space and query time for KDE data-structures, and the details of this are presented in Section 5.

### 1.3 RELATED WORK

There is a large body of work on sublinear time KDE for low dimensional spaces, which includes the classical work on Fast Gauss Transform (Greengard & Strain, 1991) and other tree based hierarchical partitioning methods (Gray & Moore, 2001; 2003; Yang et al., 2003; Lee et al., 2005; Ram et al., 2009; Gan & Bailis, 2017). For high dimensional spaces ($d = \Omega(\log n)$), sublinear time algorithms beating random sampling for various kernels such as Gaussian and polynomial were obtained by a recent sequence of works based on implementing importance sampling via LSH (Charikar & Siminelakis, 2017; Backurs et al., 2018; Charikar et al., 2020). These importance sampling based procedures had $1/\epsilon^2$ dependence on $\epsilon$ in query complexity, and works based on discrepancy theory and randomized space partitioning (Phillips & Tai, 2020; Charikar et al., 2024) achieve a $1/\epsilon$ dependence. Recent works (Siminelakis et al., 2019; Backurs et al., 2019) address scalability issues of the original approach of Charikar & Siminelakis (2017) and obtain practical improvements on real world datasets.

## 2 PRELIMINARIES

The goal of this section is to present basic notation and assumptions used throughout the paper, as well as preliminary concepts and tools regarding KDE and $(c, r)$-ANN data-structures.

**Notation.** We denote $\exp_a(b) = a^b$ and let $[n] = \{1, \ldots, n\}$ for any natural number $n$.

### 2.1 BASIC SETUP

We now present standard assumptions on parameters as part of problem setup. We first define the Gaussian Kernel.

**Definition 4** (Gaussian Kernel). $K(\mathbf{p}, \mathbf{q}) = e^{-\frac{\log(1/\mu)}{2}\|\mathbf{p}-\mathbf{q}\|^2}$. We use this version of the Gaussian Kernel because an instance with general Gaussian kernel with arbitrary bandwidth parameter as in Equation 2 can be reduced to this version using standard scaling techniques (Refer to Charikar et al. (2020, Assumption 1 in Section 5)).

**Definition 5** (Setup). The approximation factor is $\epsilon = \Omega(1/\operatorname{polylog} n)$ and $\mu^* = n^{-\Theta(1)}$ and dimension $d = \tilde{O}(1)$ (see Charikar et al. (2020, Remark 1)). We assume we know a baseline approximation $\mu$ satisfying $\mu^* \leq \mu \leq 4\mu^*$ (see Charikar et al. (2020, Remark 3)).

Note that $\mu^* = n^{-\Theta(1)}$ is the interesting regime for this problem because for $\mu^* = n^{-\omega(1)}$ under the Orthogonal Vectors Conjecture (Rubinstein, 2018), the problem cannot be solved faster than $n^{1-o(1)}$ using space $n^{2-o(1)}$ (Charikar & Siminelakis, 2019), and for larger values $\mu^* = n^{-o(1)}$ random sampling solves the problem in $n^{o(1)}/\epsilon^2$ time and space.

### 2.2 $(c, r)$-ANN ON THE SPHERE

We now present the definition of the $(c, r)$-ANN problem.

**Definition 6** (The $(c, r)$-ANN problem). Given an $n$-point dataset $\mathcal{P} \in \mathbb{R}^d$, the goal is to preprocess $\mathcal{P}$ to answer the following queries. Given a query point $\mathbf{q} \in X$ such that there exists a data point within distance $r$ from $\mathbf{q}$, return a data point within distance $cr$ from $\mathbf{q}$.

The $(c, r)$-ANN problem on the sphere is defined similarly, with the assumption that the dataset $\mathcal{P}$ contains points that lie on the unit sphere. We now state the asymmetric LSH of Andoni et al. (2017) as described in Razenshteyn (2017) for the $(c, r)$-ANN problem on the sphere.

**Theorem 7** ($(c, r)$-ANN parameters). *Razenshteyn (2017, Theorem 2.8.1) Let $\epsilon_0 > 0$ be a fixed constant. For every $c > 1$, $\frac{1}{\log\log n} \leq r = o(1)$, and for every $\rho_q, \rho_s \geq 0$, such that $cr \leq 2 - \epsilon_0$ and*

$$(c^2 + 1) \cdot \sqrt{\rho_q} + (c^2 - 1) \cdot \sqrt{\rho_s} \geq 2c \tag{8}$$

*there exists a data-structure for $(c, r)$-ANN on a unit sphere $S^{d-1} \subset \mathbb{R}^d$ where $d = n^{o(1)}$ for a set of size $n$, with space $n^{1+\rho_s+o(1)}$, query time $n^{\rho_q+o(1)}$ and success probability $1 - \frac{1}{n^{10}}$.*

We make two important remarks about this data-structure. The first, this data-structure is data-independent (see Razenshteyn (2017)). Roughly, this feature makes the data-structure more straightforward compared to data-independent ones, as they do not make any use of (or assumptions on) the dataset for preprocessing. This simpler setting allows usually for a cleaner analysis (see for example the data-dependent/independent settings in Andoni et al. (2017); Charikar et al. (2020)).

Secondly, we elaborate briefly on the query procedure Algorithm 4 of this data-structure. The basic object underlying this ANN data-structure is a tree, where each inner node contains random Gaussian vectors, and the leaves contain subsets of the processed input dataset. Importantly, querying the data-structure follows multiple paths in the tree, which are determined by the correlation of the query with the Gaussian vectors stored in the inner tree nodes. Every traversed path leaves to a leaf that contains multiple points from the original dataset. We often say that the union of all points in the reached leaves *collide* with the query. We elaborate on the data-structure's query/preprocessing algorithms as well as the parameter setting for the theorem above in Appendix A.

We now state properties of a key reduction to reduce general instances to the unit sphere.

**Lemma 8.** *There exists a reduction from $(c, r)$ -ANN problem over the $\ell_2$ for $n$-point dataset in $\mathbb{R}^d$, to $(c, r')$-ANN on the sphere problem over the $\ell_2$ distance for $n$-points on the unit sphere in $\mathbb{R}^{d+1}$ where $r' = \frac{r}{R}$ in which* all *the points are mapped to a sphere of radius $R = r \cdot \log \log n$ and then scaled by $R$ into the unit sphere. The pairwise distances between points are preserved up to scaling by $R$ and an additive factor $O(1/(r\sqrt{\log \log n}))$. This incurs an $n^{o(1)}$ query time overhead.*

Note that the reduction from the lemma above (Lemma 8) allows for recovering the *original* $(c, r)$-ANN problem, hence the points recovered by the $(c, r')$-ANN on the sphere are converted to points in the original dataset. This standard reduction was previously used in Razenshteyn (2017); Andoni et al. (2017), and we provide more details about it in Appendix A.1.

## 3 FRAMEWORK FOR NON-ADAPTIVE KDE

In this section, we introduce and generalize the framework of Charikar et al. (2020) which "reduces" KDE to an ANN problem we refer to as the Level-$j$ Recovery. In the following, we present the KDE data-structure in terms of a data-structure for the Level-$j$ Recovery problem.

Throughout the rest of the section, we assume that we are given an approximation parameter $\epsilon$ and some baseline approximation $\mu$ as in the setup (Definition 5) and Gaussian kernel (Definition 4). The first concept is that of geometric level sets.

**Definition 9** (Geometric level sets). Let $J = \lceil \log_2 \frac{1}{\mu} \rceil$. For any $j \in [J]$ and a query $\mathbf{q}$, define the level set:
$$\mathcal{L}_j^{\mathbf{q}} := \left\{ \mathbf{p}_i \in \mathcal{P} : K(\mathbf{p}_i, \mathbf{q}) \in (2^{-j}, 2^{-J+1}] \right\}.$$

This induces corresponding distance levels: $r_j := \max \left\{ r : f(r) \in (2^{-j}, 2^{-j+1}] \right\}$. Here $f(r) := K(\mathbf{p}, \mathbf{p}')$ for $r = \|\mathbf{p} - \mathbf{p}'\|$. Also define $\mathcal{L}_{J+1}^{\mathbf{q}} := \mathcal{P} \setminus \bigcup_{j \in [J]} \mathcal{L}_j^{\mathbf{q}}$.

Similarly to Charikar et al. (2020) we will sub-sample the dataset $\mathcal{P}$ at different geometric rates for each $j \in [J]$, with the goal of recovering points from $\mathcal{L}_j^{\mathbf{q}}$ given the query $\mathbf{q}$, and thus we need the following definition of a subsampled dataset and the Level-$j$ Recovery problem.

**Definition 10.** For $j \in [J + 1]$, let $\mathcal{P}_j$ be the dataset achieved by sampling $\mathcal{P}$ at rate $p_j := \min(\frac{1}{2^j n\mu}, 1)$ for $j \leq J$ and $p_{J+1} = \frac{1}{n}$. Let $m_j := \frac{1}{2^j \mu}$ be the expected size of $\mathcal{P}_j$.

**Definition 11** (Level-$j$ Recovery data-structure). Given the sample $\mathcal{P}_j$ and a point $\mathbf{q}$, recover all points in $\mathcal{L}_j^{\mathbf{q}}$ from $\mathcal{P}_j$ with probability at least $1 - \frac{1}{n^{10}}$. A data-structure for the Level-$j$ Recovery problem is parameterized by its space denoted $\mathsf{space}(j)$ and its query time denoted $\mathsf{query}(j)$.

**Remark 12.** In the paper, we will construct data-structures for the sample $\mathcal{P}_j$ for $j \in [J]$. We ignore the last sampled set, $\mathcal{P}_{J+1}$, which contains, in expectation, only a constant number of points in expectation, and hence requires constant query time and space.

As in Charikar et al. (2020), the main technical work is dedicated to constructing efficient data-structures for the Level-$j$ Recovery $\mathcal{D}_j$, which we use in the algorithms below. We use our data-structure for $j$'s that are within a *range $j \in [c_0 J, (1 - c_1)J]$* where $c_0, c_1$ can be set to any arbitrarily

small constant, our data structure and details of it are in Section 4). Assuming the *nice* range $c_0, c_1$ is fixed, for $x < c_0, x > 1 - c_1$ we use the data-structure from Charikar et al. (2020) for the Level-$j$ Recovery problem for these small $j$'s. We provide the formal statement about the guarantee of this data-structure in Appendix B.2.

**Data-structure Description.** We now describe the preprocessing and query procedures for the KDE data-structure based on those described in Charikar et al. (2020, Algorithms 1,2).

---

**Algorithm 1:** KDE PREPROCESS

> **Input:** dataset $\mathcal{P}$, precision parameter $\epsilon$, baseline approximation $\mu$ as in Definition 5, small
> constants $c_0, c_1 \in (0, 1/2)$
> 1   $K \leftarrow \dfrac{C \log n}{\epsilon^2} \cdot \mu^{-o(1)}$.
> 2   **for** $K$ *times* **do**
> 3     **for** $j \leftarrow 1$ **to** $J$ **do**
> 4       $\mathcal{P}_j \leftarrow$ subsample of $\mathcal{P}$ at rate $p_j$ from Definition 10.
> 5       **if** $j < c_0 \cdot J$ *or* $j > (1 - c_1)J$ **then**
> 6         Preprocess $\mathcal{P}_j$ using the data-structure from Lemma 27.
> 7       **else**
> 8         Preprocess $\mathcal{P}_j$ using our new data-structure $\mathcal{D}_j$ from Lemma 15.
> 9     Store a sampling of $\mathcal{P}$ with probability $1/n$.

---

**Algorithm 2:** KDE QUERY

> **Input:** Query $\mathbf{q}$ (the repetition parameter $K$ is as in Algorithm 1).
> **Output:** A $1 \pm \epsilon$ estimate for $\mu^*$.
> 1   **for** $K$ *times* **do**
> 2     **for** $j \leftarrow 1$ **to** $J + 1$ **do**
> 3       Query the Level-$j$ Recovery data-structure on $\mathbf{q}$ to recover points from $\mathcal{L}_j^{\mathbf{q}}$, for the
>         relevant repetition.
> 4     $S \leftarrow$ the set of all recovered points for the relevant repetition.
> 5     Calculate the estimate $Z \leftarrow \sum_{j \in [J]} \sum_{\mathbf{p} \in S \cap \mathcal{L}_j^{\mathbf{q}}} \frac{K(\mathbf{p}, \mathbf{q})}{p_j}$ (where $p_j$ is defined in Definition
>       10) for the relevant repetition.
> 6   **return** the average of the estimations $Z$ across all repetitions.

---

**Query Time and Space Requirement.** We now state the theorem from Charikar et al. (2020) which parametrizes the space used by Algorithm 1 and time of Algorithm 2.

**Theorem 13.** *Charikar et al. (2020, Theorems 15, 22) For Gaussian kernel $K(\mathbf{p}, \mathbf{q})$, precision parameter $\epsilon$ and baseline approximation $\mu$ as in the setup (Definition 5), and assuming that for any $j \in [J]$ there exists a data-structure $\mathcal{D}_j$ for the Level-$j$ Recovery problem with expected query time* query($j$) *and expected space requirement* space($j$), *then there exists a KDE data-structure that supports $(1 \pm \epsilon)$-multiplicative factor approximation to the KDE value with the following parameters:*

- *KDE preprocessing (Algorithm 1) uses expected space $\widetilde{O}\left(\epsilon^{-2} \cdot \max_{j \in [J]} \text{space}(j)\right)$.*

- *KDE query (Algorithm 2) runs in expected time $\widetilde{O}\left(\epsilon^{-2} \cdot \max_{j \in [J]} \text{query}(j)\right)$.*

We cite the relevant claims justifying the above in Appendix B.3. Next we derive expressions for query($j$) and space($j$) for our data-strucutre $\mathcal{D}_j$ we use in Algorithms 1 and 2 for Gaussian Kernel.

## 4   DATA-STRUCTURE FOR THE LEVEL-$j$ RECOVERY PROBLEM

We now present our data-structure $\mathcal{D}_j$ for Level-$j$ Recovery. Notice that for $r \in [0, \sqrt{2}]$, $(1/\mu)^{-r^2/2} \in [\mu, 1]$, and so we can focus our attention on $r$'s within that range (as for other values

of $r$, the contribution of points from these distances to the kernel value of any queried point amounts to $o(1/\mu)$). Using the Gaussian Kernel in Definition 9 gives the distance level $r_j = \sqrt{2j/J}$ for each $j \in [J]$. We also use the distance scale $x_j = j/J$, hence $r_j = \sqrt{2x_j}$.

**Setting up the $(c, r)$-ANN problem on the sphere.** We will use a data-structure for $(c, r)$-ANN to solve Level-$j$ Recovery. Our dataset will be the sample $\mathcal{P}_j$ (see Definition 10) with expected size $m_j = \exp_{1/\mu}(1 - x_j)$. The near distance will be $r = \sqrt{2x_j}$ and far distance $cr = \sqrt{2}$, thus $c = \sqrt{1/x_j}$. We use the data-structure from Theorem 7 for $(c, r)$-ANN problem on the sphere, thus to use this first we transform our points to lie on the unit sphere Lemma 8 (see Appendix A.1 for full details). This reduction incurs certain considerations, the most important of which is that in the following we make the assumption that $j$ lies within the *nice* range $[c_0 J, (1 - c_1)J]$ for some small constants $c_0, c_1 \in (0, 1/2)$. In this range, $j = O(J)$ and the size of the dataset is $m_j = (1/\mu)^{O(1)}$. These simplify our calculations, and have little influence since $c_0, c_1$ are chosen arbitrarily small.

**The query/space requirements of our data-structure.** The data-structure for the $(c, r)$-ANN we use is as per Theorem 7. Our data-structure $\mathcal{D}_j$ will build on top of this data-structure as follows. The preprocessing will remain the same, and so is the space requirement. For the query procedure we apply the query procedure of the data-structure for $(c, r)$-ANN problem on the sphere (Algorithm 4) but go over *all* points in the leaves reached by the ANN-query procedure. We analyze the expected number of points from level sets $\mathcal{L}_i^{\mathbf{q}}$ for $i \neq j$ that appear in the leaves of the data-structure for a given query $\mathbf{q}$. We formally analyze it in the our main technical lemma in the appendix, Lemma 31, which gives a data-structure for the Level-$j$ Recovery based on the data-structure for $(c, r)$-ANN problem on the sphere from Theorem 7 for any choice of $\rho_q, \rho_s$ that satisfies Equation (8).

**Restricting the space requirement.** Since the data-structure for the $(c, r)$-ANN problem on the sphere from Theorem 7 is parameterized by $\rho_q, \rho_s$, we need to explain the specific choice of these parameters for our setting of the Level-$j$ Recovery data-structure. For any $\delta \geq 0$, we choose to set the parameters so that the space requirement of the Level-$j$ Recovery data-structure is bounded by $\exp_{1/\mu}(1 + \delta + o(1))$. This choice enforces a constraint on the space exponent $\rho_s$:

$$\exp_{m_j}(1 + \rho_s + o(1)) \leq \exp_{1/\mu}(1 + \delta + o(1)) \tag{9}$$

and as a result, it also enforces a constraint on the query exponent $\rho_q$ by the ANN-tradeoff in Equation (8). These constrains splits the range of $x_j \in [0, 1]$ (correspondingly, $j \in [J]$) into two regimes, where the threshold between them is $\theta(\delta)$ which is the upper bound on the regimes of $x_j$ at which Equation (9) holds. In the first regime, we call the constant query distance scales, one can set $\rho_q \geq 0$ (which implies that the query time for the ANN problem becomes constant), since the smallest space that supports this does not exceed the query time. For the second regime we call the polynomial query distance scales, the space is upper bounded to not exceed our restriction, which enforces constrains on the allowed values $\rho_q$ (which implies that the query time for the ANN problem becomes polynomial). For further discussion refer to Appendix C, this is summarized as follows.

**Definition 14** (Thresholds for Query/Space Exponents). For $\delta \geq 0$ and $x \in [0, 1]$ we let:

Threshold function: $\theta(\delta) = \frac{1}{2}\left(\sqrt{(\delta + 1)(\delta + 9)} - (\delta + 3)\right)$

Space and Query Exponents Bounds (to be used in Lemma 15):

$$\rho_s(\delta, x) = \begin{cases} \frac{4x}{(1-x)^2} & \text{if } x \leq \theta(\delta) \\ \frac{\delta + x}{1 - x} & \text{if } x > \theta(\delta) \end{cases} \quad , \quad \rho_q(\delta, x) = \begin{cases} 0 & \text{if } x \leq \theta(\delta) \\ \left(\frac{2\sqrt{x} - \sqrt{(1-x)(\delta+x)}}{1+x}\right)^2 & \text{if } x > \theta(\delta) \end{cases}$$

**Putting everything together.** Our data-structure for Level-$j$ Recovery is obtained by instantiating Lemma 31 with the parameters chosen above. Its properties are in the following lemma, and its proof is in Appendix C.

**Lemma 15.** *For $\delta \geq 0$, small constants $c_0, c_1 \in (0, 1/2)$, $j \in [c_0 J, (1 - c_1)J]$ (where $x_j = j/J$), $\rho_q(\delta, x)$ from Definition 14, the data-structure $\mathcal{D}_j$ for the Level-$j$ Recovery problem with preprocess and query procedures from Algorithms 5 and 6 (found in Appendix C) has (expected) query time at most: $\exp_{1/\mu}(\xi(\delta, x_j) + o(1))$ and (expected) space at most: $\exp_{1/\mu}(1 + \delta + o(1))$ where*

$$\xi(\delta, x) = \min_{\rho \geq \rho_q(\delta, x)} \max_{y \in [x, 1]} (y - x) + (1 - x)\left(\rho - \frac{x}{y(1-x)^2}\left(\frac{y-x}{\sqrt{x}} - (y-1)\sqrt{\rho}\right)^2\right) \tag{10}$$

## 5    KDE DATA-STRUCTURE TRADEOFFS

In this section, we use the data-structure $\mathcal{D}_j$ from Lemma 15 to construct a KDE data-structure. Since our data-structure is parameterized by $\delta$ such that its space requirement is $(1/\mu)^{1+\delta+o(1)}$, we can also plug different value of $\delta$ and get a space-query tradeoff for our KDE data-structure as we do in Figure 1.

**Theorem 16.** *For any $\delta \geq 0$, precision parameter $\epsilon$ and baseline approximation $\mu$ as in the setup (Definition 5), there exists a KDE data-structure for the Gaussian Kernel (see Definition 4) that supports $(1 \pm \epsilon)$-multiplicative factor approximation to the Kernel value, in expected query time at most $\widetilde{O}\left(\epsilon^{-2} \cdot \exp_{1/\mu}\left(\xi(\delta) + o(1)\right)\right)$ time, and expected space at most at most $\widetilde{O}\left(\epsilon^{-2} \cdot \exp_{1/\mu}\left(1 + \delta + o(1)\right)\right)$ where $\xi(\delta) = \max_{x \in [0,1]} \xi(\delta, x)$ for $\xi(\delta, x)$ from Equation* (10).

The above theorem follows by plugging the parameters of the relevant data-structures into Theorem 13 (see proof in Appendix D). We also show two consequences of Theorem 16 which follow by numerical evaluations. These highlight the best query time achievable in polynomial space, and the query time achievable with linear space (see proof in Appendix D).

**Theorem 17.** *For any precision parameter $\epsilon$ and baseline approximation $\mu$ as in the setup (Definition 5), there exists a KDE data-structure for the Gaussian Kernel that allows for approximating $\mu^* := K(\mathcal{P}, \mathbf{q})$ up to $(1 \pm \epsilon)$ multiplicative factor, in the following two regimes of expected query time and space:*

- *Query time at most: $\exp_{1/\mu}\left(0.05 + o(1)\right)$ and space at most: $\exp_{1/\mu}\left(4.1 + o(1)\right)$*

- *Query time at most: $\exp_{1/\mu}\left(0.1865 + o(1)\right)$ and space at most: $\exp_{1/\mu}\left(1 + o(1)\right)$*

The query exponent Charikar et al. (2020) get for the data-independent LSH setting is $0.25$, [2], and in general they get $0.173$, both cases with essentially linear space. Our main result could be interpreted as significantly improving the query time exponent over their main result, with the caveat that their space requirement is only $1/\mu$ (compared to $1/\mu^{4.15}$ for us), or from the perspective that even within the same space constraints, when $\delta = 0$, our query exponent gets quite close to their main result with a much simpler analysis. Finally, we computed numerically the values of the query exponent $\xi(\delta, x)$ and the KDE query exponent $\xi(\delta)$, and plot these in Figure 1. This plot demonstrates the plateau of the KDE query time $\xi(\delta)$ at around $0.05$, and that for $\delta \approx 3.15$ increasing the allowed space does not yield improved query time. This limitation had been discussed in Section 1.2. We discuss these plots further in Appendix D.

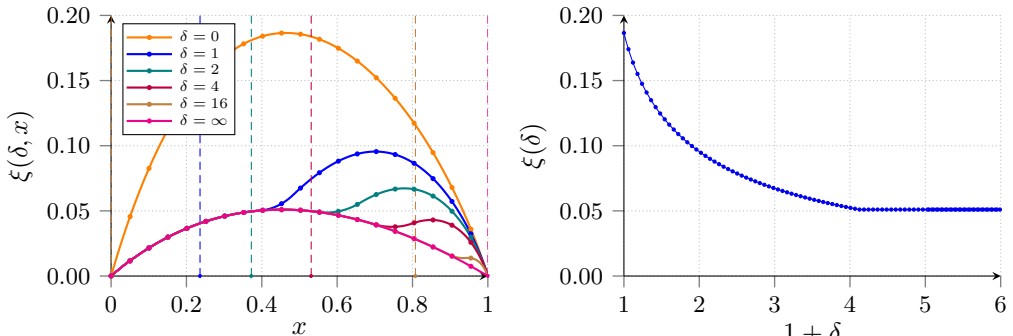

Figure 1: Left: $\xi(\delta, x)$ from Equation (10), $\delta \in \{0, 1, 2, 4, 16, \infty\}$ (dashed verticals at $x = \theta(\delta)$). Right: KDE space exponent $(1 + \delta)$ vs. KDE query exponent $\xi(\delta) = \max_{x \in [0,1]} \xi(\delta, x)$.

---

[2]We mention that one could further recover this result using our data-structure by equating the space and query exponent, i.e., forcing a symmetric setting of parameters.

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

# A SPHERICAL $(c, r)$-ANN DATA-STRUCTURE FROM RAZENSHTEYN (2017)

The data-structure for solving the $(c, r)$-ANN problem on the sphere from Razenshteyn (2017, Section 2.4) is central to our work, and we begin by defining the data-structure and stating its guarantees. The data-structure is parameterized by two parameters $\eta_s, \eta_q$ governing the space-query time trade-off (which are related tho $\rho_q, \rho_s$ in Theorem 7, as in Remark 24). Given a dataset $\mathcal{P} \subset S^{d-1}$ of $n$

points on unit sphere in $d$-dimensions, preprocessing procedure is as follows:

---

**Algorithm 3:** ANN (ON THE SPHERE) PREPROCESS

---

**Input:** dataset $\mathcal{P}$, parameters $T, K, \eta_s, \eta_q$

1 Initialize a tree with $K + 1$ levels (from 0'th level to $K$'th) and an upper bound of $T$ of the out degree. There are $T^K$ nodes in the $K$'th level.

2 Let $v_0$ denote the root, and $\mathcal{L}_v$ the path (excluding $v_0$) to any node $v$.

3 Store a random Gaussian vector $\mathbf{z}_v$ for each node $v$ except the root.

4 Define:
$$P_v = \{\mathbf{p} \in P : \forall v' \in \mathcal{L}_v, \ \langle \mathbf{z}_{v'}, \mathbf{p} \rangle \geq \eta\}$$
Every leaf $v$ at level $K$ stores the subset $P_v$ explicitly.

5 Recursively build the tree as follows - For a given node $v$, sample $T$ Gaussian vectors $\mathbf{g}_1, ..., \mathbf{g}_T \sim \mathcal{N}(0, 1)^d$. Then for every $i$ such that $\{\mathbf{p} \in P_v : \langle \mathbf{g}_i, \mathbf{p} \rangle \geq \eta_s\}$ is non-empty, we create a new child $v'$ with $\mathbf{z}_{v'} = \mathbf{g}_i$, and recursively process $v'$.

---

After preprocessing the dataset, when we are given a query we use the following procedure to return an approximate near neighbor.

---

**Algorithm 4:** ANN (ON THE SPHERE) QUERY

---

**Input:** Tree from Algorithm 3, parameters $T, K, \eta_s, \eta_q$, query $q \in S^{d-1}$

1 To answer a query $\mathbf{q} \in S^{d-1}$, we start from the root $v_0$ and traverse the tree.

2 Upon traversing node $v$, consider every child of $v$ for which $\langle \mathbf{z}_v, \mathbf{q} \rangle \geq \eta_q$ where $\eta_q > 0$, and proceed recursively.

3 If leaf node reached, return the first point with distance $\leq cr$ to $q$. ▷ See Remark 18

---

**Remark 18.** For the ANN problem, it suffices to return the first point encountered at distance $< cr$ from the queried point. In our use of this algorithm we assume that *all* points in the leaves reached by the query algorithm are returned.

To state the space and query time of the above data-structure, we will need the following notation, which will be useful for describing the properties of the LSH function.

**Definition 19.** For any $\rho \geq 0$ and $\mathbf{z} \in S^{d-1}$ let $F(\rho)$ be defined as, $F(\rho) = \Pr_{\mathbf{z} \sim \mathcal{N}(0,1)^d}[\langle \mathbf{z}, \mathbf{u} \rangle \geq \rho]$ and for any $\sigma \geq 0$ and $\mathbf{u} \in S^{d-1}$ such that $\|\mathbf{u} - \mathbf{z}\|_2 = s$ let $G(s, \rho, \sigma)$ be defined as, $G(s, \rho, \sigma) = \Pr_{\mathbf{z} \sim \mathcal{N}(0,1)^d}[\langle \mathbf{z}, \mathbf{u} \rangle \geq \rho$ and $\langle \mathbf{z}, \mathbf{v} \rangle \geq \sigma]$.

We now state the success probability, space and query time of the preprocess and query procedures of Algorithms 3 and 4. For the stating these claims we assume that there exists $\mathbf{p} \in P$ for query $\mathbf{q}$ with $\|\mathbf{p} - \mathbf{q}\| \leq r$.

**Claim 20** (Success probability). Razenshteyn (2017, Lemma 2.8.4) For any $N \geq 0$, if $T \geq \frac{10 \log n}{G(r, \eta_s, \eta_q)}$ then the probability that there is at least one leaf in the data structure created by Algorithm 3 where $\mathbf{p}, \mathbf{q}$ collide during Algorithm 4 is at least $1 - \frac{1}{n^{10}}$[3].

**Claim 21** (Space). Razenshteyn (2017, Lemma 2.8.5) The expected space required for the data-structure created by Algorithm 3 is at most: $n^{1+o(1)} \cdot K \cdot (T \cdot F(\eta_s))^K$.

**Claim 22** (Query time). Razenshteyn (2017, Lemma 2.8.6) If $TF(\eta_q) \geq 3$ then the expected run-time of Algorithm 4 is at most: $n^{o(1)} \cdot \left(T \cdot (T \cdot F(\eta_q))^K + n \cdot (T \cdot G(cr, \eta_s, \eta_q))^K\right)$.

For the above claim, the proof actually shows the following: the expected query time spent going down the tree in Algorithm 4, without scanning the leaves is $n^{o(1)} \cdot \left(T \cdot (T \cdot F(\eta_q))^K\right)$. Moreover, the expected number of points scanned at the leaves reached is $n^{1+o(1)} \cdot (T \cdot G(cr, \eta_s, \eta_q))^K$. The number of points scanned is always at most one more than the number of far points, i.e., lying a distance greater than $cr$ from $\mathbf{q}$, that reached the same leaf. Additionally, we present the following corollary, implicit in Razenshteyn (2017, Lemma 2.8.6)

**Claim 23.** For any query $\mathbf{q}$ and $\mathbf{p} \in \mathcal{P}$ such that $\|\mathbf{p} - \mathbf{q}\| \geq t$ and each leaf $\ell$ in the tree constructed in Algorithm 3, the probability that both $\mathbf{p}$ and a query $\mathbf{q}$ end up in $\ell$ is at most: $(G(t, \eta_s, \eta_q))^K$.

---

[3]This is a slight variation of the original claim from Razenshteyn (2017) which trivially follows from its original proof.

**Remark 24** (Parameter setting for Theorem 7). The result claimed by theorem Theorem 7 is achieved by instantiating Claims 20, 21, 22 with the following parameter setting, for $\rho_q \geq 0$ (see also Razenshteyn (2017, Sections 2.8.4, 2.8.5)):

1. $K = \sqrt{\ln n}$

2. $\sqrt{\tau_s} = \frac{\alpha(r)\beta(cr) - \alpha(cr)\beta(r)\sqrt{\rho_q}}{\alpha(r) - \alpha(cr)}, \qquad \sqrt{\tau_q} = \frac{\beta(cr) - \beta(r)\sqrt{\rho_q}}{\alpha(r) - \alpha(cr)}$

3. $\sqrt{\rho_s} = \frac{\beta(r)\beta(cr) - (1 - \alpha(r)\alpha(cr))\sqrt{\rho_q}}{\alpha(r) - \alpha(cr)}$

4. $\eta_s = \sqrt{2\tau_s}\,\ln^{1/4} n, \quad \eta_q = \sqrt{2\tau_q}\,\ln^{1/4} n.$

5. $G(t, \eta_s, \eta_q)^K = n^{-\frac{\tau_s + \tau_q - 2\alpha(t)\cdot\sqrt{\tau_s \tau_q}}{\beta(t)^2}} \cdot e^{\pm O(\sqrt{\log n}\cdot\log\log n)}$ for $t$ such that $r \leq t \leq cr$.

6. $T = \frac{10\log n}{G(r, \eta_s, \eta_q)} \leq e^{O\left(\sqrt{\log n}\cdot(\log(\log n))^4\right)} \leq n^{o(1)}.$

7. $F(\eta_q)^K = n \cdot G(cr, \eta_s, \eta_q)^K.$

where for $0 \leq s \leq 2$ we use $\alpha(s) = 1 - \frac{s^2}{2}$ to denote the *cosine* angle between two points on the unit sphere with distance $s$ between them, and $\beta(s) = \sqrt{1 - \alpha(s)^2}$ for $0 \leq s \leq 2$ for the sine of the same angle.

## A.1 ANN TO ANN ON THE SPHERE (PROOF SKETCH FOR LEMMA 8)

In order to apply the data-structure for the ANN on the sphere presented in Appendix A to general datasets not necessarily on the sphere, we will need a reduction. In particular we will need a reduction of $(c, r)$-ANN problem for $n$-point dataset in $\mathbb{R}^d$, to $(c, r')$-ANN problem for $n$-point dataset on the unit sphere in $\mathbb{R}^d$ where $r' = \frac{1}{\log\log n}$. This reduction is taken almost verbatim from Razenshteyn (2017, Section 2.5) (we assume that all points are normalized so that $r = 1$):

1. We reduce the dimension to $d = \log^{1+o(1)} n$ by using the dimension reduction lemma of Johnson-Lindenstrauss. This step introduces multiplicative distortion $1 \pm o(1)$ for pairwise distances, which is acceptable for us.

2. Next, we reduce the diameter of the dataset to $O((\log\log n)^{1/4})$. This can be done by partitioning the dataset using LSH family from Datar et al. (2004) and querying the part, where the query belongs. We need to repeat this procedure $n^{o(1)}$ times to get high probability of success.

3. Finally, we reduce the problem to the unit sphere case with $r = \frac{1}{\log\log n}$. This reduction can be found in Valiant (2015).

These steps imply that we can, with no asymptotic costs to the other parameters, consider the use of data-structure for the $(c, r)$-ANN problem on the sphere (up to $n^{o(1)}$ factor in query time and $o(1)$ in the other parameters). An implicit property of the reduction is critical for us and is captured by the following claim:

**Claim 25** (Implicit in the proof of Corollary 3.4 from Andoni et al. (2017)). Let $\mathcal{P} \subset \mathbb{R}^d$ be a set of points contained within a ball of radius $D$ centered at the origin, such that for all $x \in \mathcal{P}$, $\|x\|_2 \leq D$. Let $R$ be a real parameter such that $R \gg D$. Define a mapping $\phi : \mathcal{P} \to \mathbb{R}^{d+1}$ by first lifting a point $x$ to the hyperplane at height $R$ via $\pi(x) = (x, R)$, and then projecting it radially onto the sphere of radius $R$:

$$\phi(x) = \frac{R}{\|\pi(x)\|_2}\pi(x) \quad \text{and} \quad \pi(x) = \frac{R}{\sqrt{\|x\|_2^2 + R^2}}(x, R)$$

This mapping has the following properties:

1. Mapping to the sphere: $\forall x \in \mathcal{P}$, the image $\phi(x)$ lies on the sphere of radius $R$ in $\mathbb{R}^{d+1}$.

2. Single point displacement: the distance between $\pi(x)$ and the projected point $\phi(x)$ is bounded by: $\|\phi(x) - \pi(x)\|_2 \leq \frac{\|x\|_2^2}{2R}$.

3. Pairwise-distance distortion: For any two points $x, y \in \mathcal{P}$, the distance between their images is no greater than the original distance: $\|\phi(x) - \phi(y)\|_2 \leq \frac{D^2}{R} + \|x - y\|_2$.

We use the above claim with $D = O((\log \log n)^{1/4})$ (which we get from the second step in the reduction) and $R = r \cdot D^2 \sqrt{\log \log n} = r \cdot \log \log n$. The mapping $\phi(x)$ maps points to a sphere of radius $R$, with only a small distortion, and in order to reduce the points to the unit sphere we scale all points by $1/R$. Thus we obtain a critical *linearity* property of the reduction essential for our analysis, which is Lemma 8.

# B  Statements from Charikar et al. (2020)

## B.1  Level-Sets Size Bounds

The following claim about the level sets $\mathcal{L}_j^{\mathbf{q}}$ (see Definition 9) bounds the size of level sets $\mathcal{L}_j^{\mathbf{q}}$ using a simple Markov bound.

**Claim 26.** Charikar et al. (2020, Lemma 20) $|\mathcal{L}_j^{\mathbf{q}}| \leq 2^j n \mu^* \leq 2^j n \mu$ for all $j \in [J]$.

## B.2  Previous Level-$j$ Recovery data-structure

We recall the parameters of the data-structure for the Level-$j$ Recovery that was considered in that paper.

**Lemma 27.** *Charikar et al. (2020, Theorems 15, 22) For the Gaussian kernel $K$ and every $j \in [J]$, given the sample $\mathcal{P}_j$, and a query $\mathbf{q}$, there exists a data-structure $\mathcal{D}_j$ for the Level-$j$ Recovery that uses the following query and space bounds (where $x_j = \frac{j}{J}$):*

- *(expected) query time at most:* $\exp_{1/\mu}\left(x_j\left(1 - x_j\right)\left(1 + o(1)\right)\right)$

- *(expected) space at most:* $\min\left(n \cdot \exp_{1/\mu}\left(x_j\left(1 - x_j\right)\left(1 + o(1)\right)\right), \exp_{1/\mu}\left(1 + o(1)\right)\right)$

The above implies that for $j$ smaller than some arbitrarily small constant $\tau$, one gets query and space requirements that are arbitrarily small. For the space bound we remark that in any case, our new data-structure in Section 4 has space requirement of at least $1/\mu$, since we ignore the first term of the minimization in our discussions.

## B.3  Estimator Accuracy for Theorem 13

In this subsection we cite the claim from Charikar et al. (2020) that prove that the KDE data structure from Algorithms 1 and 2 supports $(1 \pm \epsilon)$ approximation for $\epsilon$ as in the Setup (Definition Theorem 5).

The first claim argues that, conditioned upon the success of the ANN data-structure, argues that the estimator is unbiased.

**Claim 28** (Charikar et al. (2020), Claim 24). Let $\mu^* \in (0, 1)$, $\mu \geq \mu^*$, $\epsilon \in (\mu^{10}, 1)$, $\mathbf{q} \in \mathbb{R}^d$. Assume that for any data-set and for any of the $K$ repetition (see the definition of $K$ in Algorithm 1), the data structure $\mathcal{D}_j$ constructed in Algorithm 1 is able to solve Level-$j$ Recovery (Definition 11) with probability at least $1 - \frac{1}{n^{10}}$, the estimator $Z$ for any repetition constructed in Algorithm 2 satisfies the following:
$$(1 - n^{-9})n \cdot \mu^* \leq \mathbb{E}[Z] \leq n \cdot \mu^*$$

We note that Charikar et al. (2020, Claim 24), as written in the reference, uses a previous claim about the recovering probability of the data-structure they construct, but is essentially oblivious to it's inner working, and in fact it works for any data-structure with good enough recovery probability. The next claim from Charikar et al. (2020) argues that running Algorithms 1 and 2 with a constant factor approximation to $\mu^*$ suffices to obtain an accurate estimate of $\mu^*$.

**Claim 29.** Charikar et al. (2020, Claim 25) For every $\mu^* \in (0, 1)$, every $\epsilon \in (\mu^{10}, 1)$, every $\mathbf{q} \in \mathbb{R}^d$, using estimators $Z$, for every repetition constructed in Algorithm 1 where $\mu/4 \leq \mu^* \leq \mu$, one can output a $(1 \pm \epsilon)$-factor approximation to $\mu^*$.

## C    DEFERRED PROOFS FROM SECTION 4

Our main goal in this section is to prove Lemma 15. Before that we need to prove various intermediate helper lemmas. Our first lemma is a general statement about the $(c, r)$-ANN data-structure from Section A. It quantifies the probability that points from intermediate bands $t \in [r, cr]$ appear in the set of points in the leaves that is gathered in Algorithm 4.

**Lemma 30.** *Let $\mathcal{P}$ be a dataset of size $n$ that contains points on the unit sphere, which is preprocessed into a data-structure for $(c, r)$-ANN problem on the sphere with a query parameter $\rho_q$. Let $t \in [r, cr]$. Given a query $\mathbf{q}$ and a point $\mathbf{p} \in \mathcal{P}$ at distance at least $t$ from $\mathbf{q}$, the probability that $\mathbf{p}, \mathbf{q}$ collide in at least one leaf is at most:*

$$
\exp_n \left( \rho_q - \frac{\left( c(t^2 - r^2)\sqrt{1 - \frac{c^2 r^2}{4}} - (t^2 - c^2 r^2)\sqrt{1 - \frac{r^2}{4}}\sqrt{\rho_q} \right)^2}{(c^2 - 1)^2 r^2 t^2 \left( 1 - \frac{t^2}{4} \right)} \right)
$$

*Proof.* We are interested in the probability that there is at least one leaf in the data structure constructed in Algorithm 3 where $\mathbf{p}, \mathbf{q}$ collide. By Claim 23, for each point $\mathbf{p}$, at distance $t$, and each leaf $\ell$ in the tree, the probability that both $\mathbf{p}$ and $\mathbf{q}$ end up in $\ell$ is at most: $(G(t, \eta_s, \eta_q))^K$. Since there are $T^K$ leaves, by union bound, the probability that $\mathbf{p}$ appears in the leaves reached for query $\mathbf{q}$ is at most: $T^K \cdot (G(t, \eta_s, \eta_q))^K$. Then we get:

$$
T^K \cdot (G(t, \eta_s, \eta_q))^K = n^{\left( \rho_q - \frac{((\alpha(r) - \alpha(t))\beta(cr) - (\alpha(cr) - \alpha(t))\beta(r)\sqrt{\rho_q})^2}{\beta^2(t)(\alpha(r) - \alpha(cr))^2} \right) + o(1)}
$$

$$
= n^{\left( \rho_q - \frac{\left( c(t^2 - r^2)\sqrt{1 - \frac{c^2 r^2}{4}} - (t^2 - c^2 r^2)\sqrt{1 - \frac{r^2}{4}}\sqrt{\rho_q} \right)^2}{(c^2 - 1)^2 r^2 t^2 \left( 1 - \frac{t^2}{4} \right)} \right) + o(1)}
$$

The bound in the claim follows by using the parameters and the definitions of $\alpha(\cdot), \beta(\cdot)$ in Remark 24. $\qquad \square$

**Algorithms for the data-structure $\mathcal{D}_j$ for the Level-$j$ Recovery.** Next, we give a formal description of the data-structure we use for the Level-$j$ Recovery problem, used to preprocess $\mathcal{P}_j$ (of size $m_j$, see Definition 10). We emphasize that the underlying data-structure we use is essentially the data-structure for the $(c, r)$-ANN problem on the sphere from above (with relevant parameters $c, r$), with a small but crucial difference in the query procedure. We also use the sphere reduction from Lemma 8 which was discussed in Appendix A.1.

---

**Algorithm 5:** $\mathcal{D}_j$ PREPROCESS

    **Input:** Dataset $\mathcal{P}_j$ (of size $m_j$, see Definition 10), parameters $\rho_q, \rho_s$ (and letting $x_j = j/J$)

1  Reduce from $(\sqrt{1/x_j}, \sqrt{2x_j})$-ANN to $(\sqrt{1/x_j}, \sqrt{2x_j}/R)$-ANN on the sphere where
    $R = \sqrt{2x_j} \cdot \log\log((1/\mu)^{1-x_j})$ (see Lemma 8).

2  Use the data-structure for ANN on the sphere from Theorem 7 with parameters $\rho_s$ and $\rho_q$.

---

**Algorithm 6:** $\mathcal{D}_j$ QUERY

    **Input:** A query $\mathbf{q}$, a preprocessed ANN on the sphere data-structure for $\mathcal{P}_j$ via Algorithm 5.

1  Use the sphere reduction from Lemma 8 to convert $\mathbf{q}$ to a valid query $\mathbf{q}'$ on the sphere.

2  Run the query algorithm of the ANN on the sphere Algorithm 4 on $\mathbf{q}'$.

3  Scan *all* returned points to retrieve all points (on the sphere) from the leaves at distance
    $r_j/R$ from $\mathbf{q}'$ (see Definition 9).

4  Return the corresponding points from $\mathcal{P}_j$ to the points recovered in the previous step.

---

**Overview of our analysis.**    After describing the algorithm, we provide a general statement about the query time and space requirement of the data-structure. Note that the space is exactly as in the data-structure for the $(c, r)$-ANN problem on the sphere, but the query time is quite different, and we now explain its derivation. Recall that during when querying $\mathbf{q}$, the query algorithm of the $(c, r)$-ANN on the sphere in Algorithm 4, ends up with multiple points that collided with the query $\mathbf{q}$ (see Appendix A for more details).

The query time of the ANN structure accounts for the number of *far* points, i.e., at distance most bigger than $cr$ from the query. Since our goal is to recover the points in $\mathcal{L}_j^{\mathbf{q}}$ that survives the sampling for $\mathcal{P}_j$, we would need to scan *all* the points reached by the data-structure for $\mathbf{q}$. However, from the density bounds of Claim 26 we can bound how many points from $\mathcal{L}_i^{\mathbf{q}}$ for $i \neq j$ both "survived" the sampling step and also collide with a point in $\mathcal{L}_j^{\mathbf{q}}$ in expectation. Using properties of the ANN scheme on the sphere from Appendix A, we bound the probability they collide with $\mathbf{q}$ in the ANN structure. These considerations make the query time of our data-structure, up to polylog factors, the maximum over other level sets $i \neq j$ of the number of expected points that are in the sample $\mathcal{P}_j$ and collide with $\mathbf{q}$ during the query (which is potentially bigger than the query time for the ANN data-structure).

**Lemma 31.** *For $\delta \geq 0$, $j \in [c_0 J, (1 - c_1)J]$ (for arbitrary small constants $c_0, c_1$), $\rho_q, \rho_s \geq 0$ that satisfy Equation* (8)*, there exists a data-structure $\mathcal{D}_j$ for the Level-$j$ Recovery problem that uses preprocess and query procedures from Algorithms 5 and 6 and has the following properties (where $x_j = j/J$):*

- *(expected) query time at most:* $\exp_{1/\mu}\left(\gamma(\rho_q, x_j) + o(1)\right)$

- *(expected) space at most:* $\exp_{1/\mu}\left(1 + \rho_s + o(1)\right)$

*where:*

$$\gamma(\rho, x) = \max_{y \in [x, 1]} (y - x) + (1 - x)\left(\rho - \frac{x}{y(1-x)^2}\left(\frac{y-x}{\sqrt{x}} - (y-1)\sqrt{\rho}\right)^2\right) \qquad (11)$$

*Proof.* In the proof we will use the normalize index $x_j = j/J$ instead of $j \in [J]$ and correspondingly $x_i = i/J$ for $i \in [J] \setminus \{j\}$. Our goal is to construct a data-structure that will enable us to recover all points in $\mathcal{L}_j^{\mathbf{q}}$ in the sampled set $\mathcal{P}_j$, or equivalently, all points at distance at most $r_j$ from the query $\mathbf{q}$. Our overall strategy will be to construct a ANN on the sphere data-structure from Theorem 7 (See Appendix A) on the sampled set $\mathcal{P}_j$, whose size is size is $m_j = 1/(2^j \mu) = (1/\mu)^{1-x_j}$ (see Definition 10).

Within the sampled dataset, we focus our attention on points at distances $r$ from $\mathbf{q}$ such that $r \in [0, \sqrt{2}]$, as otherwise, their contribution to the kernel value amounts to $o(1/\mu)$ (see Definition 4). Since our points of interest in the dataset have $r$ within that range, we aim to construct a data-structure for the $(c, r)$-ANN problem on the sphere where $c = \sqrt{2}/r_j$ and $r = r_j$ (such that the close points are at distance $r_j$ and the far at distance $\sqrt{2}$). In the following discussion, the parameters $\rho_q, \rho_s$ we use in for the ANN data-structure are unspecified (since at this point, they can be chosen arbitrarily). Since $r_j = \sqrt{2x_j}$ for the Gaussian Kernel by the definition of the Gaussian Kernel (Definition 4 and the level set radii in Definition 9), we have $c = \sqrt{1/x}$.

Now we address the number of points from different levels $\mathcal{L}_i^{\mathbf{q}}$ for $i \neq j$ are in subsampled set $\mathcal{P}_j$. By Claim 26, the size of the sampled set $\mathcal{L}_i^{\mathbf{q}}$ for each $i$ is at most $2^i n\mu = n\mu^{1-x_i}$. Since the set $\mathcal{P}_j$ is sampled at rate $p_j = 1/(2^j n\mu) = \mu^{1-x_j}/n$, the number of sampled points from levels $x_i < x_j$ is at most $O(1)$:

$$\sum_{x_i < x_j} n\mu^{1-x_i} \cdot \frac{1}{n\mu^{1-x_j}} = \sum_{x_i < x_j} \frac{1}{\mu^{x_i - x_j}} = O(1)$$

Additionally, in expectation we have at most one point from $\mathcal{L}_j^{\mathbf{q}}$ after subsampling (again by Claim 26), which we want to recover, and there is expected number of $2(1/\mu)^{x_i - x_j} = O((1/\mu)^{x_i - x_j})$ points from bands $L_i^{\mathbf{q}}$ for $i > j$ in the set $\mathcal{P}_j$ (see Definition 9). We denote the exponent of that quantity by:

$$\Phi(x_j, x_i) = x_i - x_j$$

The points from $\mathcal{L}_i^{\mathbf{q}}$ are at distance at least $r_i = \sqrt{2i/J} = \sqrt{2x_i}$ from $\mathbf{q}$, while our data-structure was constructed for $r_j = \sqrt{2x_j}$.

We now use Lemma 8 to transform the problem from a $(\sqrt{1/x_j}, \sqrt{2x_j})$-ANN problem of points in $\mathbb{R}^d$ to the $(\sqrt{1/x_j}, \sqrt{2x_j}/R)$-ANN problem on the sphere where and the points lie on a sphere $S^d$, for $R = \sqrt{2x_j} \cdot \log\log((1/\mu)^{1-x_j})$ (see also Appendix A.1 for more details). This reduction incurs a small negligible $(1/\mu)^{o(1)}$ factor to the query time. We note that since $j$ is in the *nice* regime, $x_j \in [c_0, 1 - c_1]$, this implies that the size of the dataset is $m_j = (1/\mu)^{O(1)}$, the radius for our level of interest is $r_j = O(1)$ (also $x_j = O(1)$) and the scaling $R = O(\log\log((1/\mu)^{O(1)}) = \omega(1)$.

By Lemma 8, the sphere reduction keeps pairwise distances (before scaling by $R$) the same up to a factor of $O(1/(r_j\sqrt{\log\log m_j})) = o(1)$, and the scaling changes the pairwise distances by a factor of $1/R$. Hence, we can think of the induced level-sets $\mathcal{L}_j^{\mathbf{q}}$ of the dataset (for every $j \in [J]$) *after* the reduction as containing the points that would have been contained in the corresponding level-sets $\mathcal{L}_j^{\mathbf{q}}$ *before* the reduction, with the only difference being that the distances $r_j$ are scaled to $r_j/R$ for ever $j \in [J]$. Hence, in the following we continue with the assumption that the points are on the sphere (and hence we can use the data-structure for ANN on the sphere), and keep the notation for the level sets and the query. We also use the fact that the reduction from Lemma 8 allows to recover the points from the original dataset, before the sphere reduction.

Recall that our goal is to recover the points in $\mathcal{L}_j^{\mathbf{q}}$ from $\mathcal{P}_j$. These points will collide with $\mathbf{q}$ when running the query algorithm Algorithm 4 of the data-structure for ANN on the sphere as per Claim 20 with $1 - 1/\text{poly}(n)$ success probability. For our setting, the query Algorithm 6 scans *all* points reached by the query algorithm for ANN on the sphere, and hence we must account bound the number of points from all $\mathcal{L}_i^{\mathbf{q}}$ for $i > j$ that collide with the query $\mathbf{q}$ in the data-structure for the ANN on the sphere and are therefore returned by the query algorithm (note that for $i < j$ we saw that there's only a constant amount in $\mathcal{P}_j$). Applying Claim 30 for $t = r_i/R$, $r = r_j/R$ and $c = \sqrt{2}/r_j$ (and since the dataset is of size $(1/\mu)^{1-x}$) we get that the collision probability between a query $\mathbf{q}$ and points from $\mathcal{L}_i^{\mathbf{q}}$ in the sample $\mathcal{P}_j$, is at most $(1/\mu)^{\chi(\rho_q, x_j, x_i)}$ where:

$$\chi(\rho_q, x_j, x_i) = (1-x)\left( \rho_q - \frac{\left( c(r_i^2 - r_j^2)\sqrt{1 - \frac{c^2 r_j^2}{4R^2}} - (r_i^2 - c^2 r_j^2)\sqrt{1 - \frac{r_j^2}{4R^2}}\sqrt{\rho_q} \right)^2}{(c^2 - 1)^2 r_j^2 r_i^2 \left(1 - \frac{r_i^2}{4R^2}\right)} + o(1) \right)$$

$$= (1 - x_j)\left( \rho_q - \frac{\left( \frac{x_i - x_j}{\sqrt{x_j}}\sqrt{1 - \frac{1}{2R^4}} - (x_i - 1)\sqrt{1 - \frac{x_j}{2R^4}}\sqrt{\rho_q} \right)^2}{\frac{(1-x_j)^2 x_i}{x}\left(1 - \frac{x_i}{2R^4}\right)} + o(1) \right)$$

$$= (1 - x_j)\left( \rho_q - \frac{x_j}{x_i(1-x_j)^2}\left( \frac{x_i - x_j}{\sqrt{x_j}} - (x_i - 1)\sqrt{\rho_q} \right)^2 + o(1) \right)$$

The second equality follows by plugging the values of $r_i, r_j$ and the last equality follows by putting aside $o(1)$ terms coming from the fact that $R = \omega(1)$. Hence, the expected number of points from $\mathcal{L}_i^{\mathbf{q}}$ that collide with $\mathbf{q}$ in the $\mathcal{P}_j$ is at most:

$$\mathbb{E}[\text{probability that points from } \mathcal{L}_i^{\mathbf{q}} \text{ collide with } \mathbf{q}] \leq (1/\mu)^{\Phi(x_j, x_i) + \chi(\rho_q, x_j, x_i)}$$

Now we bound the expected number of collisions with $\mathbf{q}$ in the sample $\mathcal{P}_j$ from all $i > j$ :

$$\mathbb{E}[\text{number of points in } \mathcal{L}_i^{\mathbf{q}} \cap \mathcal{P}_j, \forall i > j \text{ colliding with } \mathbf{q}] \leq \sum_{i > j} (1/\mu)^{\Phi(x,y) + \chi(\rho_q, x, y)}$$

$$\leq J \cdot \max_{i \in [j, J]} (1/\mu)^{\Phi(x_j, x_i) + \chi(\rho_q, x_j, x_i)}$$

$$\leq \widetilde{O}\left( \max_{i \in [j, J]} (1/\mu)^{\Phi(x_j, x_i) + \chi(\rho_q, x_j, x_i)} \right)$$

$$\leq (1/\mu)^{\max_{y \in [x, 1]} \Phi(x_j, y) + \chi(\rho_q, x_j, y) + o(1)}$$

and we define:

$$\gamma(\rho, x) = \max_{y \in [0,1]} \Phi(x, y) + \chi(\rho, x, y)$$

$$= \max_{y \in [0,1]} (y - x) + (1 - x) \left( \rho - \frac{x}{y(1-x)^2} \left( \frac{y - x}{\sqrt{x}} - (y - 1)\sqrt{\rho} \right)^2 \right).$$

This latter function is the function $\gamma(\rho, x)$ defined in the lemma statement. We note that the final upper bound maximization over the continuous range $y \in [0, 1]$ to account for all in-between level-set collision probabilities, as the distances between points and the query lie within a range of distances, according to the level-set definition in Definition 9 (and hence we do use the analytic behavior of the collision probability). Finally, we mention that maximizing over $y \in [0, 1]$ rather than $y \in [x, 1]$ does not decrease the maximum (and more so because since, as seen above, no $y$ such that $y < x$ would not be maximizer). □

**Explanation for Definition 14**   We elaborate on the two regimes for $j \in [J]$ (correspondingly $x_j = j/J$) when enforcing the space limit in Equation (9).

- *Constant query distance scales ($x_j \leq \theta(\delta)$).* In this regime, even if one uses maximal setting of $\rho_s = \frac{4x}{(1-x)^2}$ (which matches the lower bound enforces by plugging $\rho_q = 0$ and the value of $c$ into Equation (8)) the space requirement of the data-structure does not exceed $\exp_{1\mu}(1 + \delta + o(1))$. Hence, any setting of $\rho_q \geq 0$ could be used.

- *Polynomial query distance scales ($x > \theta(\delta)$.* in this regimes the exponent $\rho_s$ that satisfies Equation (9) must be such that $\rho_s \leq \frac{\delta + x}{1 - x}$. Plugging the upper bound on the space exponent (in order to allow as much flexibility for the choice of the query exponent) into the ANN-relation 8, gives a lower bound on $\rho_q$.

Finally, we conclude with the proof of Lemma 15

*Proof of Lemma 15.* The proof follows by instantiating Lemma 31 with specific value of $\rho_q, \rho_s$. Our choice of parameters, is to set $\rho_s = \rho_s(\delta, x_j)$ (where $x_j = j/J$), where $\rho_s(\delta, x_j)$ is defined in Definition 14 as the space exponent that guarantees that the space requirement of the data structure will be at most $(1/\mu)^{1+\delta+o(1)}$ (see also the discussion above and in Section 4). The value of $\rho_q$ is set to be the value of $\rho$ that minimizes query exponent of a Level-$j$ data-structure based on the data-structure for the $(c, r)$-ANN problem on the sphere, analyzed in Lemma 31 in Appendix C, which is $\min_{\rho \geq \rho_q(\delta, x_j)} \gamma(\rho, x_j)$ (where $\gamma$ is defined in Equation (11)). Note that this is exactly the expression for the query exponent $\xi(\delta, x)$ defined in Equation (10). Similarly to before, $\rho_q(\delta, x_j)$ is defined in Definition 14 as the smallest query exponent possible for the ANN data-structure on the sphere, with the space exponent limitation of $1 + \delta + o(1)$ (see also the discussion above and in Section 4). □

# D   DEFERRED PROOFS FROM SECTION 5

In this section, we provide the proofs for the theorems in Section 5.

*Proof of Theorem 16.* Recall that the query and space requirement of the KDE data-structure from Theorem 13 is set to the max query and space requirement for the Level-$j$ Recovery data-structure.

First we choose $c_0 = c_1 = 0.01$, which set the *nice* range to be $[0.01, 0.99]$. Then, we note that for this choice, and for $x < 0.01$ and $x > 0.99$, the query time of the data-structure from Lemma 27, gives query time of at most $\exp_{1/\mu}(0.01 + o(1))$. We also argue that the space requirement of this data-structure $\min \left( n \cdot \exp_{1/\mu} (0.01 + o(1)), \exp_{1/\mu} (1 + o(1)) \right)$ wouldn't matter, since for any $\delta \geq 0$, the space requirement of the dataset $\mathcal{D}_j$ is at least $\exp_{1/\mu} (1 + o(1))$.

For $x$ in the *nice* regime, we plug the parameters of our new new data-structure $\mathcal{D}_j$ for Level-$j$ Recovery from Lemma 15. For the space requirement, we set the parameters of this data-structure so that our data-structure uses $\exp_{1/\mu}(1 + \delta + o(1))$. For query time, we set it to be maximal over

$x \in [c_0, 1 - c_1]$ of $\xi(\delta, x)$, which is defined in Equation (10) as the exponent of the query time for the Level-$j$ Recovery data-structure. We can therefore upper bound the query time exponent by $\max_{x \in [0,1]} \xi(\delta, x)$ which is $\xi(\delta)$. Note that we can consider $x \in [0, 1]$ instead of in the *nice* range as we give an upper bound on the expected query time.

Finally, we note that from numerical evaluations, we see that $\xi(\delta) \geq 0.04$ for any $\delta$ (see also Figure 1, where the graph of $\xi(\delta)$ plateaus, and discussion below), which means that the query time in the range $x < 0.01$ or $x > 0.99$ doesn't effect the maximization. □

*Proof of Theorem 17.* The proof of this lemma is based on numerical evaluations of the expression for $\xi(\delta) = \max_{x \in [0,1]} \xi(\delta, x)$, where $\xi(\delta, x)$ is defined in Equation (10). Our first result identifies the value of $\delta$ at which

$$\max_{x > \theta(\delta)} \xi(\delta, x) \leq \max_{x \leq \theta(\delta)} \xi(\delta, x) \tag{12}$$

for $\theta(\delta)$ defined in Definition 14. At this point, the maximal value of the curve $\xi(\delta, x)$ for $x \leq \theta(\delta)$ is higher than that for $x > \theta(\delta)$. While the value of $\xi(\delta, x)$ for $x > \theta(\delta)$ decreases with $\delta$, the value of $\xi(\delta, x)$ for $x \leq \theta(\delta)$ is in fact independent of $\delta$. Hence, from the value of $\delta$ at which Equation (12) holds, the value of $\xi(\delta) = \max_{x \in [0,1]} \xi(\delta, x)$ cannot decrease further. Via numerical evaluations, we get that Equation (12) occurs at $\delta \approx 3.15$ and $\xi(\delta) \approx 0.05$. These values are also demonstrated in the left figure of Figure 1 (and discussed below). Similarly, for $\delta = 0$, one can numerically evaluate $\xi(\delta)$ (noting that $\rho_q(\delta, x)$ from Theorem 14 is 0, which simplifies the constraints for evaluations). Numerically we get that $\xi(0) = 0.1865$.

Finally, we provide a script (in Appendix D.1) for reproducing the numbers we report. The script estimates the value of $\xi(\delta)$ at $\delta = 0$ and prints the point $\delta$ at which the minimum of $\xi(\delta)$ is first achieved. □

**On the functions $\xi(\delta, x)$ and $\xi(\delta)$.** We elaborate on Figure 1 which show the following. The left figure shows the exponent function for the query exponent of our data-structure $\xi(\delta, x)$ for the level $x$ (assuming $x = j/J$, see Lemma 15 and Equation (10)). The right figure shows the query time exponent of the KDE data-structure $\xi(\delta) = \max_{x \in [0,1]} \xi(\delta, x)$ (see Theorems 13 and 16). Both of these functions are plotted in Figure 1, where the inner maximizations and minimizations are computed numerically (similarly to the script in Appendix D.1).

- The left figure in Figure 1 shows the behavior of the function $\xi(\delta, x)$ for different values of $\delta$ (as a function of $x \in [0, 1]$). The vertical lines denote the thresholds for the distance scales $\theta(\delta)$ from Definition 14 (the threshold where below one can use constant query for the underlying data-structure for ANN on the sphere. See discussion in Appendix D). The threshold and the curve for the same $\delta$ are colored in the same color.

  This figure shows (in pink for $\delta = \infty$) the *underlying curve* that every curve with $\delta > 0$ "emerges" from after passing the threshold $\theta(\delta)$. This curve is the curve one gets from using $\rho \geq 0$ in the maximization of $\xi(\delta, x)$ for the entire range of $x \in [0, 1]$, as if there is no threshold $\theta(\delta)$ (or rather, the threshold is equal to 1).

  One sees that every curve for $\delta$ diverges from the underlying curve at some point after $\theta(\delta)$. Up to a point, an increase in $\delta$ (equivalently, increasing the threshold $\theta(\delta)$), *decreases* the global maximum of the function (see for example the differences between using $\delta = 0, 1, 2$). One sees from the value of $\delta$ at which the underlying curve's maximum is the global maximum of the function (numerically, around $\delta \approx 3.15$), no increase in the value of $\delta$ can lead to a smaller query exponent $\xi(\delta, x)$, which plateaus at the maximum of the underlying curve (at around 0.05). This is demonstrated by the curves for $\delta > 4$.

- Similarly, we plot the query exponent of the KDE data-structure $\xi(\delta)$ (see Theorem 16) and present it on the right plot of Figure 1. This function takes the maximum over $x \in [0, 1]$ of $\xi(\delta, x)$, and as expected and demonstrated by the plot for $\xi(\delta, x)$, it initially decreases when $1 + \delta$ increases but then plateaus. The explanation for this phenomena lies in the cases induced by the threshold $\theta(\delta)$. As before, as $\delta$ increases, $\theta(\delta)$ approaches 1, which means that for most of the $x$'s $\rho_q(x, \delta) \geq 0$. In that case, we have that $\xi(\delta)$ is actually *independent* of $\delta$ (as in that regime $\rho_q(x, \delta) = 0$, see Definition 14) which explains why this plateau is not affected by the change in $\delta$.

## D.1 EVALUATION SCRIPT

This section discusses a python script (which can be found in the supplementary material to this paper) used to evaluate the query-exponent $\xi(\delta)$ (see Equation (10) and theorem 16) and reproduce the results of Theorem 17. The maximizations and minimizations are computed via a grid search (the grid size is configurable and can be increased to improve accuracy).

In terms of notation, the script uses $F$ instead of $\xi$ to denote the exponent function, and:

$$f(x, y, \rho) = (y - x) + (1 - x) \left( \rho - \frac{x}{y(1-x)^2} \left( \frac{y-x}{\sqrt{x}} - (y-1)\sqrt{\rho} \right)^2 \right)$$

(which is the same as the function $\xi(\delta)$ but without the optimization over $x, y, \rho$). The value of $F(\delta) = \max_{x \in [0,1]} \min_{\rho \geq \rho_q(\delta, x)} \max_{y \in [0,1]} f(x, y, \rho)$ is therefore the same as $\xi(\delta)$ (where $\rho_q(\delta, x)$ is defined in Definition 14).

We note that the script was created using ChatGPT 5 (as well as generating the plots in Figure 1).

