# OpenReview forum: "Faster Kernel Density Estimation via Hashing Based Time–Space Tradeoffs"
_ICLR.cc/2026/Conference — ICLR 2026 Conference Withdrawn Submission_

### Official Review · Reviewer_aK7V · 2025-10-28

**Soundness:** 3
**Presentation:** 3
**Contribution:** 3
**Rating:** 8
**Confidence:** 4

**Summary:**

This paper proposes a data structure that improves query time at the expense of higher space complexity than existing methods. The proposed techniques provide known query-time vs. space trade-offs for KDE. The query time improves the best-known non-adaptive KDE bound and nearly matches the best-known bound.

**Strengths:**

S1. This paper presents strong theoretical contributions to a significant problem in theoretical ML.

S2. This paper introduces novel and sound techniques that improve upon existing methods.

S3. Although this paper contains dense mathematical results, its organization is clear and can be followed without delving into mathematical details.

**Weaknesses:**

W1. This paper is purely theoretical, with no empirical study. I suspect whether such a contribution is of enough interest to the ICLR community.

W2. Although the analysis procedure is claimed to be simpler, the query time does not match the best-known one ($1/\mu^{0.173}$ vs. $1/\mu^{0.1865}$) with the same space complexity.

**Questions:**

Q1. I am not sure which kernel functions are supported by the proposed technique. Gaussian kernel only or general kernels that can be reduced to LSH and ANN computations?

---

> ### Author Response · Authors · 2025-11-18
>
> Thank you for taking the time to review our paper - we appreciate your thoughtful comments and questions. Below, we address the points you've raised:
>
> Weaknesses - As far as the simplicity of our approach is concerned, note that the data-dependent result of [Charikar, Kapralov, Nouri, Siminelakis’20] involves an intricate reduction to a linear program, which in turn is basically a discretization of an integral equation. Our approach achieves comparable results by optimising tradeoffs for data-independent ANN, which we feel is valuable because of the simplicity and the ability to produce an entire tradeoff curve. For example, if we allow $1/\mu^{1.1}$, which is fairly close to linear, we obtain a query time of $1/\mu^{0.167}$, improving even on the data-dependent result of [Charikar, Kapralov, Nouri, Siminelakis’20].
>
> Questions - Thank you for this question, it's an excellent point. While our analysis could, in principle, apply to other kernels, it does not necessarily improve upon the results of [Charikar, Kapralov, Nouri, Siminelakis’20] for kernels like the exponential kernel. The key issue is subtle: LSH and ANN structures behave differently in how they induce collision probabilities. LSH guarantees a collision profile based on a close distance $r$, while ANN structures specify behavior only at distances $r$ and $cr$, with no explicit guarantees in between. In our framework, we derive the intermediate collision profile (Lemma 30) based on the choice of $cr$, which depends on the kernel. For Gaussian kernels in the symmetric setting, this matches the Andoni–Indyk LSH profile, letting us recover the result of [Charikar, Kapralov, Nouri, Siminelakis’20], while for exponential kernels, it does not. We view it as an interesting open direction to achieve improved tradeoffs for other kernels beyond the Gaussian setting.
>
> We hope our responses help clarify the points raised and assist in your assessment of our work. We're happy to answer any follow-up questions or address any additional concerns you may have.

---

### Official Review · Reviewer_EBXM · 2025-10-29

**Soundness:** 3
**Presentation:** 3
**Contribution:** 3
**Rating:** 6
**Confidence:** 4

**Summary:**

The paper studies the following problem.
Given a point set $P$ in $\mathbb{R}^d$ and a query $q$ in $\mathbb{R}^d$, we would like to compute the kernel density estimation $(1/\lvert P \rvert)\sum_{p\in P} \exp(-\lVert p-q \rVert^2)$.
However, the exact computation requires a linear scan over the entire dataset which takes $n$ time and hence is prohibitive.
Therefore, the question is if we allow approximation up to a $1\pm \epsilon$ multiplicative factor, can we reduce the query time?
Previous result Charikar et al. (2020) showed that one can construct a data structure that its query time is $1/\mu^{0.173+o(1)}$ and its space complexity is $1/\mu^{1+o(1)}$ where $\mu$ is the minimum value of the ground truth kernel density estimation we are interested in.
The authors provide a novel construction of data structures that allows a tradeoff between query time and space complexity.
More precisely, the authors construct a data structure that allows the tradeoff between the setup of query time being $1/\mu^{0.05+o(1)}$ and space complexity being $1/\mu^{4.15+o(1)}$ and the setup of query time being $1/\mu^{0.1865+o(1)}$ and space complexity being $1/\mu^{1+o(1)}$.

The main idea is to use locality sensitive hashing (LSH) and approximate nearest neighbor search (ANN).
First, one can set $J = \log (1/\mu)$ and construct $J$ sets that each set is a sampled subset of $P$ at different rates $p_j$ for $j = 1,2,\dots,J$.
For each sampled subset $P_j$, maintain a data structure that uses LSH and ANN to allow faster query time for returning points that are within an annulus of $q$, $L_j$.
Finally, return the sum $\sum_{j = 1}^J \sum_{p \in L_j} (1/p_j) e^{-\lVert p-q \rVert^2}$.

**Strengths:**

- The presentation is generally clear.
The authors manage to explain the problem definition and the approach clearly.
Readers of different backgrounds should be able to understand the idea of the result.

- The result shows an interesting spectrum on the tradeoff between query time and space complexity.
I believe the result can provide some insights in the field.

**Weaknesses:**

- Though the general presentation is good, the authors may want to provide more explanation on the novelties of the approach and result.
The authors attempted to point out the novelties in the technique overview such as the use of asymmetric ANN.
More highlights on the difference between the previous approach and the current approach helps readers to fully grasp the key technical improvement.

**Questions:**

- Section 1.1: When discussing the results between the previous work and the current work, it may be helpful to make a table to show the comparison clearly.

- Line 258: Is $X$ $\mathbb{R}^d$?

- What is the challenge of extending the result for other kernels?

---

> ### Author Response · Authors · 2025-11-18
>
> Thank you for taking the time to review our paper - we appreciate your thoughtful comments and questions. Below, we address the points you've raised:
>
> Weaknesses -
> Thank you for your comment. We would like to highlight that the novelty of our work is twofold. First, we observe that in the framework of Charikar et al., the worst-case query time and the worst-case space complexity are driven by different recovery subproblems. Second, we show that asymmetric ANN data structures can be used to align these costs so that both the query time and the space complexity are dominated by the same recovery subproblem. This yields a substantial improvement in query time while keeping the space usage low.
> This leads to our main result, that to the best of our knowledge is the first to allow one to tradeoff space vs query time for KDE data-structures. Previously it was not known how to reduce the query time at the expense of polynomially increasing the space budget. Another important novelty is that our query time of $1/\mu^{0.1865}$ with linear space is achieved by data-independent techniques and improves upon the previous best data-independent bound of $1/\mu^{0.25}$. The bound of  $1/\mu^{0.1865}$ is close to the best data dependent bound of $1/\mu^{0.173}$ but with a much simpler analysis.
> Lastly, techniques recover the current best-known query exponent bound, namely 0.173, with space exponent 1.066, and therefore improve the best known query time 0.173 exponent with less than 7% higher space exponent and with a much simpler analysis.
>
> Questions -
> Thank you for these questions. Regarding the first two points - we’ll add that to the final version of the paper. For the last point, it's an excellent question. While our analysis could, in principle, apply to other kernels, it does not necessarily improve upon the results of [Charikar, Kapralov, Nouri, Siminelakis’20] for kernels like the exponential kernel. The key issue is subtle: LSH and ANN structures behave differently in how they induce collision probabilities. LSH guarantees a collision profile based on a close distance $r$, while ANN structures specify behavior only at distances $r$ and $cr$, with no explicit guarantees in between. In our framework, we derive the intermediate collision profile (Lemma 30) based on the choice of $cr$, which depends on the kernel. For Gaussian kernels in the symmetric setting, this matches the Andoni–Indyk LSH profile, letting us recover the result of [Charikar, Kapralov, Nouri, Siminelakis’20], while for exponential kernels, it does not. We view it as an interesting open direction to achieve improved tradeoffs for other kernels beyond the Gaussian setting.
>
> We hope our responses help clarify the points raised and assist in your assessment of our work. We're happy to answer any follow-up questions or address any additional concerns you may have.

---

### Official Review · Reviewer_gXvi · 2025-10-29

**Soundness:** 2
**Presentation:** 2
**Contribution:** 1
**Rating:** 2
**Confidence:** 4

**Summary:**

This paper presents a theoretical analysis of the space–time tradeoff for kernel density estimation (KDE) using hashing-based data structures. The work builds upon the framework of Charikar et al. (FOCS 2020), which reduced KDE to a density-constrained approximate nearest neighbor (ANN) problem. The authors propose replacing symmetric LSH with asymmetric LSH (Andoni et al., 2017) that was proposed to control the space-time trade-off of ANN to obtain the space-time trade-off for KDE, improving query exponents (e.g., $1/\mu^{0.05}$) at the cost of significantly increased space (e.g., $1/\mu^{4.15}$).

**Strengths:**

- Provides a clean integration of asymmetric LSH into the KDE framework of Charikar et al. (2020).

- Theoretical exposition follows the structure of the prior work, and the derivations appear internally consistent.

- The paper establishes a general tradeoff function ξ(δ) between query time and space, which could be useful for benchmarking future approaches.

**Weaknesses:**

**W1. Marginal Novelty / Lack of Theoretical Depth**

The main novelty lies in reusing the well-known data-independent asymmetric LSH time–space tradeoff (Equation 5) and substituting it into the existing reduction by Charikar et al. (2020). The result primarily reproduces a direct consequence of known ANN theory, with no fundamentally new KDE insights or analysis beyond reparameterization. When evaluating the paper as a theoretical work, I feel that it reads as a straightforward application rather than a conceptual advancement.

The improvement in query exponent (from $1/\mu^{0.173}$
 to
$1/\mu^{0.05}$) comes at the cost of extremely high space ( $1/\mu^{4.15}$), which is impractical and makes the improvement largely theoretical. In the balanced regime (δ=0), the improvement over Charikar et al. (2020) is only marginal (from 0.173 to 0.1865) with no empirical validation or deeper interpretive insight.

**W2. Unclear Role of Constants $c_0, c_1$**

The paper restricts analysis to levels $j \in [c_0 J, (1- c_1) J]$ but provides no intuition for why these constants are necessary or how they influence the resulting tradeoff (L387–L390). This omission obscures the motivation for the “nice range” assumption and whether it affects the theoretical results.

**W3. Heavy Dependence on Prior Work (Charikar et al. 2020)**

The exposition assumes the reader has read and understood Charikar et al. (2020). Several definitions, assumptions, and remarks (e.g., Assumption 1, Remark 3, and various sampling arguments) are reused verbatim or near-verbatim, making this paper hard to parse independently. Some text sections (e.g., L052–053, L249–252) are almost identical to those in Charikar et al., suggesting insufficient self-containment and a lack of clarity for new readers.

**W4. Notation and Presentation Issues**

The paper introduces new notation in Page 3 (e.g., distance scales $x_j$ such that $\mu^{x_j} \approx 2^{-j}$) but then reverts to Charikar’s notation later in Section 3. The inconsistent reuse of notation and symbols significantly reduces readability and makes it difficult to follow how variables such as $x_j, y, J$ interact across sections.

**Typos:**

- L305, 2^{-J + 1} should be 2^{-j + 1}

- strucutre

- Recoveryis

**Questions:**

Could you please address the raised weakness above and the questions below?

**Q1. Under what assumptions does $K(p, q) = \mu^{\|p - q\|}$ hold (L107)?**

If this is derived from the Gaussian kernel, then $\mu = e^{-2\sigma^2}$. How can the parameter $\sigma$  of the kernel function depend on the approximation or computed value of $\mu$? Please clarify this mapping.

**Q2. Why is it necessary to retrieve all points at distance scale $x$ (L127, Remark 3, Definition 11)?**

Charikar et al. only sample points and estimate contributions probabilistically, whereas exact recovery of all points may require time proportional to output size $|L_j \bigcup P_j|$, which undermines the sublinear query claim.

**Q3. What are the roles of $c_0, c_1$ in restricting the range $j \in [c_0 J, (1- c_1) J]$?**

If they only simplify asymptotics, please justify why results do not depend critically on these constants.

---

> ### Author Response · Authors · 2025-11-18
>
> Thank you for taking the time to review our paper - we appreciate your thoughtful comments and questions. Below, we address the points you've raised:
>
> Weaknesses -
>
> W1 - Thank you for your comment. We would like to highlight that the novelty of our work is twofold. First, we observe that in the framework of Charikar et al., the worst-case query time and the worst-case space complexity are driven by different recovery subproblems. Second, we show that asymmetric ANN data structures can be used to align these costs so that both the query time and the space complexity are dominated by the same recovery subproblem. This yields a substantial improvement in query time while keeping the space usage low.
> As a result, we achieve what is, to the best of our knowledge, the first space vs query tradeoff for KDE data-structures, previously it was not known how to reduce the query time at the expense of polynomially increasing the space budget. Furthermore we improve upon the best known query time at the expense of only $10%$ improvement in the space budget. Another important novelty is that our query time of $1/\mu^{0.1865}$ is achieved by data-independent techniques and improves upon the previous best data-independent bound of $1/\mu^{0.25}$, and is close to the best data dependent bound of $1/\mu^{0.173}$ but with a much simpler analysis.
> Lastly, another insight that follows from our work is that unlike ANN, we believe achieving an arbitrarily small query time at the expense of arbitrarily large polynomial space is not possible for KDE with existing techniques, please refer to section 1.2 page 4 for an extensive discussion.
>
> W2 - Thank you for this comment. As you point out in the “Questions” part of the review, these constants are used to simplify the asymptotics. We address their influence in L964-968 in the submitted version, where we choose both of them to be 0.01, and argue that the query/space requirements from these bands does not dominate. In fact, we could have also taken $c_0,c_1$ to be asymptotically small, as long as the radii that are induced by this choice satisfy the lower bound on the radius in theorem 7. Taking an arbitrary small constant is intended to simplify these edge-cases. In the final version of the paper, we will remark the role of these constants (and why they can essentially be ignored).
>
> W3 - Thank you for your suggestions, we do agree that we can make our paper self-contained and will do so in the final version of the paper.
>
> W4 - We believe that the current notation is justified, because while the actual data structure uses a discrete collection of sampling levels (indexed by $j$), the analysis is most naturally  viewed in terms of a continuous variable $x$. Of course, there is a place where these two views meet, namely in Lemma 31 (and the corresponding discussion in section 4), but this is the only such place.
> Typos - Thank you for pointing this out. We will fix that in the final version of the paper.
>
> Questions -
>
> Q1 - Thank you for the question. This is a standard mapping used in previous works on KDE for simplification of the analysis. As referred to on line 106, we take this from the work of  [Charikar, Kapralov, Nouri, Siminelakis’20], which we clarify here again - Suppose the given kernel function to us is $K(p,q) = e^{-|p-q|_2^2/2\sigma^2}$, then let the scaling factor $s= \sqrt{1/(\sigma^2\log(1/\mu))}$. Now for every point $p\in P \cup \{q\}$ let $p’ = sp$, thus $K(p,q) = (1/\mu)^{|p’-q’|_2^2/2}$. Thus working with the scaled points $p’ = sp$ for all $p\in P \cup \{q\}$ we can without loss of generality assume our kernel function has the specified form. We will add these details in the final version of the paper to ensure it is self-contained.
>
> Q2 - Thank you for the question. Note that we need to recover all the points at distance scale $x$ in the subsampled dataset, and after subsampling there is at most 1 point in expectation that will survive and needs to be recovered.
>
> Q3 - See response to W4.
>
> We hope our responses help clarify the points raised and assist in your assessment of our work. We're happy to answer any follow-up questions or address any additional concerns you may have.

---

> ### Comment · Reviewer_gXvi · 2025-11-19
>
> Thank for your feedback.
>
> Unfortunately, I still lean on rejection.
>
> As I evaluate this submission as a theoretical work, I find that the contribution does not yet rise to the level of a substantial conceptual or technical advancement. I find that the work achievement is an application of the space-time trade-off LSH results on the Charikar et al. framework, as the framework is built on the near neighbor search problem.
>
> The work would be significantly strengthened if the authors could demonstrate practical implications, e.g., empirical results showing how space–time trade-off LSH performs in accelerating KDE in real settings, or a clearer quantitative comparison that highlights genuine benefits over existing baselines. Such results would help justify the relevance and usefulness of the theoretical framework.
>
> I found the claim that the analysis is “simple” is also not fully convincing. The approach relies on the space–time trade-off framework for data-independent LSH, which itself depends on non-trivial and fairly intricate theoretical machinery.

---

### Official Review · Reviewer_BvtH · 2025-11-13

**Soundness:** 2
**Presentation:** 2
**Contribution:** 2
**Rating:** 2
**Confidence:** 5

**Summary:**

This paper proposes techniques to accelerate kernel sum estimation with LSH, an important theory problem in machine learning.

### Background
The seminal work in this area is the "hashing-based estimator" (HBE) framework by Charikar and Siminelakis at FOCS'17, which at a high level computes the kernel sum by (1) hashing all of the points in the dataset into LSH buckets, (2) hashing the query into the same buckets, (3) computing the kernel values for all values in the buckets where the query lands, and (4) repeating this process several times to refine the estimate. The intuition is that since LSH sends nearby points to the same buckets, and kernel values are large only for nearby points, we can get most of the mass in the kernel sum by sampling from the LSH buckets.

There have been several improvements to the original HBE work, initially focused on improving the space requirements by sampling ("Space and time efficient KDE" at NeurIPS'19) and getting the scheme to work in practice ("Rehashing kernel evaluation" at ICML'19), with a later paper that extends the scheme to more general classes of kernels ("Multi-resolution hashing" at FOCS'20). One unavoidable source of inefficiency with the HBE idea is that for radial kernels, we may have points with similar kernel contributions go to different LSH buckets (e.g., because they are on opposite sides of the query - imagine a situation where the query is the midpoint between two data clusters). This leads to redundant kernel value calculations. The most recent progress on this method (Charikar 2020), addresses this issue by counting the number of points in spherical shells around the query. This newer method (1) uniformly downsamples the dataset into subsets of exponentially-smaller sizes - one for each spherical shell radius, (2) creates an LSH index for each subset, and (3) at query-time, looks through each bucket to find the number of points in the shell radius.

### Contribution of this paper
The critical design choices in the KDE algorithm by Charikar et. al. are the partitioning + subsampling algorithm and the construction of the LSH index for each subset. Charikar et. al. consider two LSHs - a symmetric LSH which yields $\mu^{-0.25}$ query time and a data-dependent LSH which yields $\mu^{-0.173}$ query time but is considerably harder to analyze.

This paper modifies the algorithm by using an asymmetric LSH function, which induces a space-vs-query time tradeoff. When the space complexity is set to be the same as previous algorithms, this paper has $\mu^{-0.1865}$ query time. When the space is allowed to increase dramatically (to $\mu^{-4.15}$!), this paper has $\mu^{-0.05}$ query time.

**Strengths:**

1. The problem is important. Kernel sum estimation is intimately related to important practical applications such as efficient attention design, partition function estimation, and density estimation For example, given an improved algorithm for KDE one can immediately obtain an improved algorithm for transformer inference.
2. It is an interesting that the well-known space-time tradeoffs in (Andoni 2017) result in a similar tradeoff for KDE

**Weaknesses:**

1. The space complexity of $\mu^{4.15}$ is extremely high. To contextualize this number, the main contribution of *"Space and Time Efficient Kernel Density Estimation in High Dimensions"* in NeurIPS 2019 was to reduce the space complexity of HBE from $\mu^{-1.5}$ to $\mu^{-0.5}$.
2. This paper is a reasonably-interesting composition of two well-known techniques - the KDE algorithm from Charikar et. al. and the space-optimal LSHs from Andoni et. al. It would significantly add to the theory value if there were some modification / improvement to either method.
3. The presentation of the algorithm could be significantly improved. Right now, the paper seems to present two separate algorithms, but these are really just hyperparameter configurations of the "Asymmetric LSH + Charikar2020" idea. It might be better to introduce the tradeoff first, and then show how the tradeoff can reproduce algorithms. It would also be very helpful to have diagrams showing how the algorithm works (e.g., showing spherical annuli with various downsampling rates). The presentation of the theorems / proofs can also be cleaned up a lot, since in many cases the analysis is inherited from Charikar 2020 and/or Andoni 2017.
4. Many of the results in the paper are obtained via numerical evaluations. While this does not invalidate the results, it does make the analysis less elegant / insightful and more complicated.

**Questions:**

Aside from addressing the weaknesses, I would like to understand:
1. Have you tried any strategies to mitigate the space complexity? It seems like this comes pretty directly from the near-neighbor space requirements of the asymmetric LSH that is used. However, the KDE problem is subtly different because we can typically use sampling to reduce the number of points that we store in each LSH table, at the cost of a constant factor increase to the estimator variance (this is not possible for near-neighbor, since there we must return a specific point). I wonder whether you could improve the space tradeoff by doing more sampling in each LSH table.

---

> ### Author Response · Authors · 2025-11-18
>
> Thank you for taking the time to review our paper - we appreciate your thoughtful comments and questions. Below, we address the points you've raised:
>
> Weaknesses -
> 1. Thank you for your comment. Using our techniques, one can recover the current best-known query exponent bound, 0.173, with space exponent 1.066, which improves the best known query time 0.173 exponent with only 7% higher space exponent, with a much simpler analysis.
> We would like to highlight that to the best of our knowledge ours is the first result that allows one to tradeoff space vs query time for KDE data-structures, previously it was not known how to reduce the query time at the expense of polynomially increasing the space budget.
> 2. We would like to highlight that the novelty of our work is twofold. First, we observe that in the framework of Charikar et al., the worst-case query time and the worst-case space complexity are driven by different recovery subproblems. Second, we show that asymmetric ANN data structures can be used to align these costs so that both the query time and the space complexity are dominated by the same recovery subproblem. This yields a substantial improvement in query time while keeping the space usage low.
> This leads to the first query time vs space tradeoffs in the KDE problem, to the best of our knowledge. We feel that this is a compelling contribution and hope will lead to further exciting work in this area. For example, our work suggests that, unlike Nearest Neighbour Search, it is likely impossible to obtain a KDE primitive with zero query time exponent even given arbitrary polynomial space - an exciting direction for further research is to prove this lower bound formally.
> 3. Thank you for the careful reading of the paper. We will include diagrams illustrating the main concepts and improve presentation for the final version of the paper.
> 4. Thank you for your comment. The final bound that we achieve arise out of solving certain non-linear equations encapsulating the decay of the Gaussian kernel and isoperimetric properties of Euclidean space, and thus it is not clear how to solve them analytically. It is also important to note that the exponent in the previous bound of $1/\mu^{0.173}$ was also obtained using numerical simulations. Indeed, in order to obtain the bound of $1/\mu^{0.173}$ Charikar et al. use a rather intricate reduction to a linear program, which in turn approximates an integral equation. Our numerical simulations amount to simply choosing optimal points on the tradeoff curve for the underlying asymmetric LSH data structure, and our arguably conceptually simpler, which we feel is an advantage.
>
> Questions -
> Thank you very much for the interesting suggestion. We note that in each of the Level-$j$ recovery (see Definition 11, L315) problems we solve via asymmetric nearest neighbour search data structures, the expected number of nearest neighbour points that we need to recover is constant thus further subsampling will remove the points that we need to recover for KDE estimation from the sampled dataset.
>
> We hope our responses help clarify the points raised and assist in your assessment of our work. We're happy to answer any follow-up questions or address any additional concerns you may have.

---

### Note · Authors · 2025-12-04

I have read and agree with the venue's withdrawal policy on behalf of myself and my co-authors.